# p53 in AgRP neurons is required for protection against diet-induced obesity via JNK1

Mar Quiñones[1,2], Omar Al-Massadi[1,2], Cintia Folgueira[1,2], Stephan Bremser[3,4], Rosalía Gallego[5], Leonardo Torres-Leal[1], Roberta Haddad-Tóvolli[6,7], Cristina García-Caceres[8], Rene Hernandez-Bautista[1], Brian Y.H. Lam[9], Daniel Beiroa[1,2], Estrella Sanchez-Rebordelo[1,2], Ana Senra[1], Jose A. Malagon[1], Patricia Valerio[1], Marcos F. Fondevila[1], Johan Fernø[10,11], Maria M. Malagon[2,12], Raian Contreras[8], Paul Pfluger[8], Jens C. Brüning[13,14,15], Giles Yeo[9], Matthias Tschöp[8], Carlos Diéguez[1,2], Miguel López[1,2], Marc Claret[6,7], Peter Kloppenburg[3,4], Guadalupe Sabio[16] & Ruben Nogueiras[1,2]

p53 is a well-known tumor suppressor that has emerged as an important player in energy balance. However, its metabolic role in the hypothalamus remains unknown. Herein, we show that mice lacking p53 in agouti-related peptide (AgRP), but not proopiomelanocortin (POMC) or steroidogenic factor-1 (SF1) neurons, are more prone to develop diet-induced obesity and show reduced brown adipose tissue (BAT) thermogenic activity. AgRP-specific ablation of p53 resulted in increased hypothalamic c-Jun N-terminal kinase (JNK) activity before the mice developed obesity, and central inhibition of JNK reversed the obese phenotype of these mice. The overexpression of p53 in the ARC or specifically in AgRP neurons of obese mice decreased body weight and stimulated BAT thermogenesis, resulting in body weight loss. Finally, p53 in AgRP neurons regulates the ghrelin-induced food intake and body weight. Overall, our findings provide evidence that p53 in AgRP neurons is required for normal adaptations against diet-induced obesity.

[1] Department of Physiology, CIMUS, University of Santiago de Compostela-Instituto de Investigación Sanitaria, 15782 Santiago de Compostela, Spain. [2] CIBER Fisiopatología de la Obesidad y Nutrición (CIBERobn), 15706 Santiago de Compostela, Spain. [3] Biocenter, Institute for Zoology, University of Cologne, 50923 Cologne, Germany. [4] Cologne Excellence Cluster on Cellular Stress Responses in Aging-Associated Diseases, University of Cologne, 50923 Cologne, Germany. [5] Department of Morphological Sciences, School of Medicine, University of Santiago de Compostela-Instituto de Investigación Sanitaria, 15782 Santiago de Compostela, Spain. [6] Neuronal Control of Metabolism (NeuCoMe) Laboratory, Institut d'Investigacions Biomèdiques August Pi i Sunyer (IDIBAPS), 08036 Barcelona, Spain. [7] CIBER de Diabetes y Enfermedades Metabólicas Asociadas (CIBERDEM), 08036 Barcelona, Spain. [8] Institute for Diabetes and Obesity, Helmholtz Diabetes Center, Helmholtz Zentrum München, 85764 Neuherberg, Germany. [9] MRC Metabolic Diseases Unit, University of Cambridge Metabolic Research Laboratories, Wellcome Trust-MRC Institute of Metabolic Science, Addenbrooke's Hospital, Cambridge CB2 0QQ, UK. [10] KG Jebsen Center for Diabetes Research, Department of Clinical Science, University of Bergen, 5007 Bergen, Norway. [11] Hormone Laboratory, Haukeland University Hospital, 5021 Bergen, Norway. [12] Department of Cell Biology, Physiology and Immunology, Instituto Maimónides de Investigacion Biomédica de Córdoba (IMIBIC), Hospital Universitario Reina Sofia/University of Cordoba, 14004 Córdoba, Spain. [13] Department of Mouse Genetics and Metabolism, Institute for Genetics and Center for Molecular Medicine (CMMC), University of Cologne, Zülpicher Strasse 47b, 50674 Cologne, Germany. [14] Cologne Excellence Cluster on Cellular Stress Responses in Aging-Associated Diseases (CECAD), University of Cologne, Zülpicher Strasse 47b, 50674 Cologne, Germany. [15] Max Planck Institute for Neurological Research, Gleueler Strasse 50, 50931 Cologne, Germany. [16] Fundación Centro Nacional de Investigaciones Cardiovasculares Carlos III, 28029 Madrid, Spain. These authors contributed equally: Mar Quiñones, Omar Al-Massadi, Cintia Folgueira. Correspondence and requests for materials should be addressed to R.N. (email: ruben.nogueiras@usc.es)

The arcuate nucleus of the hypothalamus (ARC) is critical for the regulation of energy balance since it receives direct signals (hormones and nutrients) from the periphery[1] and comprise two neuronal populations that modulate food intake, energy expenditure, and nutrient partitioning[2]. Neurons leading to a positive energy balance express agouti-related protein (AgRP) and neuropeptide Y (NPY), while neurons leading to a negative energy balance express proopiomelanocortin (POMC)[3–5].

Data gleaned in recent years have allowed uncovering of several mechanisms implicated in obesity development and maintenance. Interestingly, many of these mechanisms have been shown to act in the hypothalamus, via altering of intercellular communication by signals such as hormones/adipokines, dysregulated nutrient sensing, and alteration in stress and inflammatory responses[2,6]. Accumulating evidence indicates that these peripheral signals act via specific hypothalamic nutrient and/or energy sensors, with deletion or activation in specific hypothalamic areas of sirtuin 1, AMP-activated protein kinase, mammalian target of rapamycin, and liver kinase B1 demonstrated to modulate whole-body energy balance[7–11]. Of note, it has recently been postulated that some of these signals/nutrient sensors may exert their effects by influencing hypothalamic proteostasis[6].

p53 is a tumor suppressor activated in response to various stimuli[12–15] and is also considered as a metabolic sensor[16]. Cumulative evidence has placed this protein, beyond its role in cancer, as a relevant metabolic regulator through its actions on glycolysis, oxidative phosphorylation, amino acid and lipid metabolism[12,15,17]. Reports showing that some p53 mutations are related with metabolic alterations suggest the clinical relevance of p53 in this regard. Patients with the Li Fraumeni syndrome, which is an autosomal dominant cancer predisposition disorder caused by some p53 mutations, have increased oxidative phosphorylation of skeletal muscle[18]. Moreover, the so-called p53 signature, an indicator of higher ovarian cancer risk in women with BRCA mutations, was inversely associated with body mass index[19]. In line with this, preclinical studies manipulating p53 in peripheral tissues showed important effects in the whole-body metabolism with pathological implications for obesity and type 2 diabetes[15,20–26]. In the hypothalamus, acetylated levels of p53 are decreased after fasting[27] and recovered after refeeding[28]. However, the pathological implications as well as the potential metabolic role of p53 in the hypothalamus remain completely unknown.

Here, we provide compelling evidence that hypothalamic p53 is a key factor in the regulation of energy balance. These metabolic actions rely specifically on AgRP neurons, since mice lacking p53 in AgRP neurons (AgRPp53 KO) are more susceptible to gain weight when fed a high fat diet (HFD), which is not the case when p53 is lacking in ARC POMC or SF1 neurons in the ventromedial nucleus of the hypothalamus (VMH). In agreement with this, mice with genetic overexpression of p53 in the ARC and more precisely in AgRP neurons are protected against diet-induced obesity (DIO). The mechanism underlying p53 actions on energy balance involves JNK1. Finally, AgRPp53 KO mice do not respond to the most potent peripheral orexigenic signal, namely, ghrelin.

## Results

**Genetic inhibition of p53 in the ARC increases weight gain.** To analyze the role of p53 in the regulation of energy balance at the central level, we started with a virogenetic approach to inhibit p53 in two different hypothalamic nuclei involved in the control of energy homeostasis, namely, ARC and VMH. For this, we injected Ad-GFP alone or a mutant Tp53RK, which acts as a dominant negative (Ad-DNp53RK) to inhibit p53 activity and

function[24] in rats fed a chow diet. The efficacy of the stereotaxic injections in the ARC was corroborated by immunostaining of GFP and DAPI (Fig. 1a) with 20% of the cells transduced after the viral injection (Supplementary Fig. 1). The functional efficiency of the viral vectors was demonstrated by decreased protein levels of phopho-p53 (active form of p53) in the ARC (Fig. 1b)[24]. The inhibition of p53 activity in the ARC increased body weight (Fig. 1c), food intake (Fig. 1d), and adiposity (Fig. 1e) after 9 days. This phenotype was associated with decreased temperature in the interscapular BAT (Fig. 1f), increased size of BAT lipid droplets (Fig. 1g), and the reduced protein levels of BAT uncoupling protein 1 (UCP1) (Fig. 1h). BAT is connected with the CNS via the sympathetic nervous system (SNS), which modulates BAT activity through the β-adrenergic receptors. We found lower mRNA expression of β-adrenergic receptors in BAT of Ad-DNp53RK rats (Fig. 1i), suggesting that inhibition of p53 in the ARC decreases SNS signaling. In addition, inhibition of p53 activity in the ARC resulted in bigger white adipocytes compared with the control group (Fig. 1j) and increased peroxisome proliferator-activated receptor gamma (PPARg) gene expression in WAT (Fig. 1k), suggesting that the inhibition of p53 activity in the ARC promotes adipogenesis.

The role of p53 in the ARC appears to be nucleus-specific, since no effect on body weight or food intake was detected when the same viral vector targeted the VMH (Supplementary Fig. 2a–c) or in mice lacking p53 in SF1 neurons (SF1p53 KO) fed either chow diet or HFD (Supplementary Fig. 2f–r).

**p53 is upregulated in the hypothalamus of diet-induced obese mice.** p53 levels are higher in the white adipose tissue[21] and in the liver[29] of obese mice. Herein, we found a significant increase in p53 protein levels in the ARC of DIO rats and in the mediobasal hypothalamus of DIO mice (Supplementary Fig. 3a, b). Consistent with this, HFD activated the p53 signaling pathway since it stimulated protein levels of the p53 downstream molecules p66-Shc and Bax in the mediobasal hypothalamus of DIO mice (Supplementary Fig. 3c). When p53 protein levels were measured in the mediobasal hypothalamus of mice fed a HFD during 2 and 4 weeks, before developing obesity, we did not find differences in p53 levels in comparison to mice fed a chow diet (Supplementary Fig. 3d, e).

**Ablation of p53 in AgRP increases susceptibility to DIO.** To shed more light about which neuronal population within the ARC is required for the metabolic actions of p53, we generated mice with selective ablation of p53 in POMC or AgRP neurons, using specific Cre transgenic mouse lines. To confirm whether mutant mice lack p53 in AgRP or POMC neurons, the tdTomato-reporter allele was introduced to allow expression of the fluorescent protein tomato in these neurons, as previously described[30]. Double immunohistochemistry revealed that p53 and tdTomato were colocalized in the ARC of control mice, and we detected p53 in approximately 80% of AgRP and POMC, while the colocalization observed in POMC-Cre or AgRP-Cre p53*loxP/loxP* mice was much lower (Supplementary Fig. 4b, c), verifying p53 deletion in hypothalamic in POMC or AgRP neurons, respectively. Moreover, FACS sorting and single-cell RNA sequencing of AgRP-eGFP neurons showed that 31 out of 45 (69%) AgRP neurons express p53 (GEO Database repository: GEO Accession: GSE92707) (Supplementary Fig. 4a).

POMCp53 KO mice showed no differences in food intake, body weight, body composition, glucose tolerance, or insulin sensitivity when compared with their control littermates after 18 weeks on chow diet or HFD (Fig. 2a–m). Similarly, AgRPp53 KO mice fed a chow diet did not show differences in body weight,

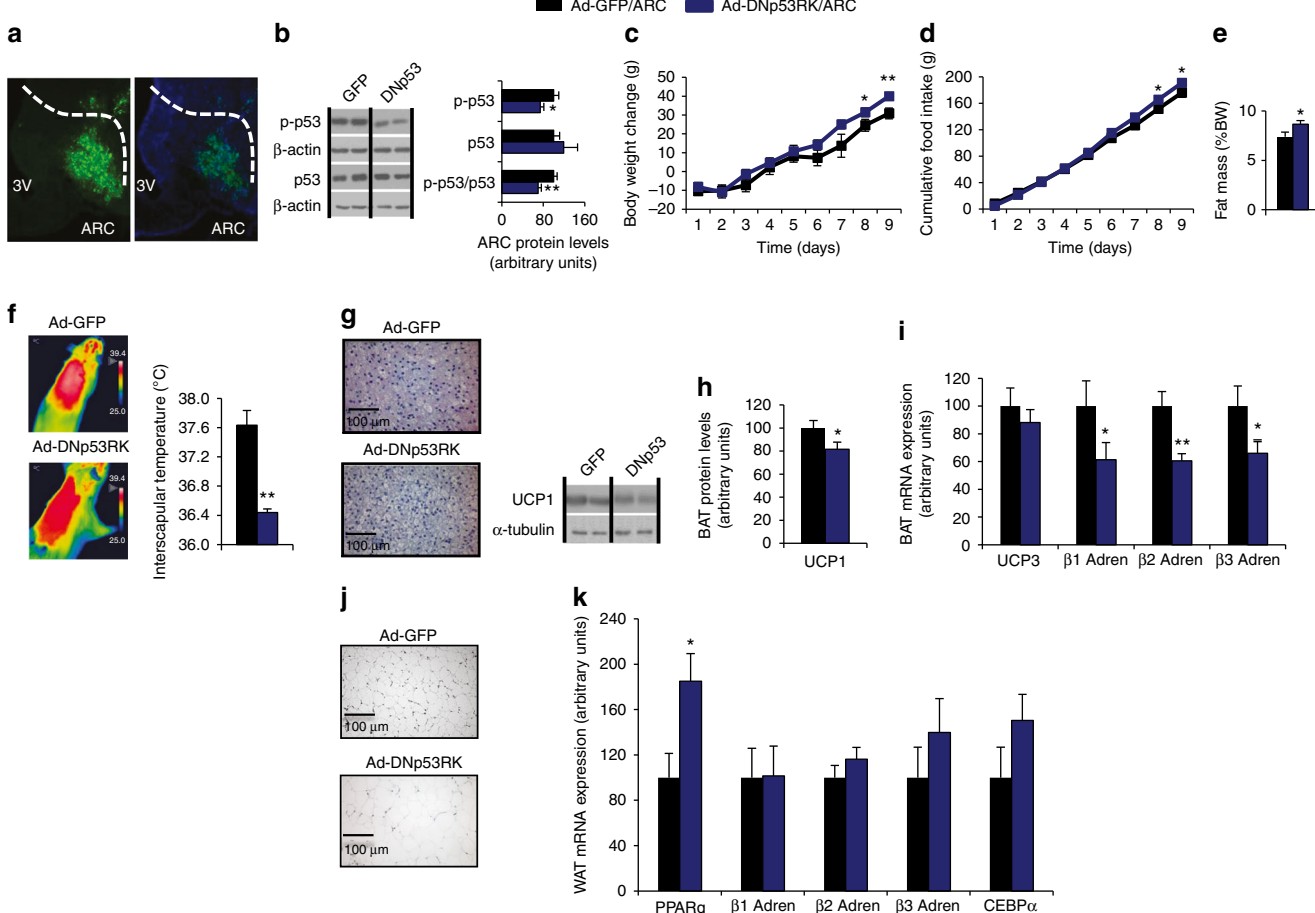

**Fig. 1** Effects of virogenetic inhibition of p53 into the ARC of rats fed chow diet on energy balance. Representative immunofluorescence showing GFP and DAPI expression in the ARC (**a**). ARC protein levels of phospho-p53 and p53 (**b**). Body weight change (**c**); cumulative food intake (**d**); fat mass (**e**); representative infrared thermal images and temperature of the BAT (**f**); representative pictures of BAT histology (**g**); BAT protein levels of UCP1 (**h**); BAT mRNA expression of UCP3, β1-adrenergic, β2-adrenergic, and β3-adrenergic receptors (**i**); representative pictures of WAT histology (**j**); and WAT mRNA expression of PPARγ, β1-adrenergic, β2-adrenergic, and β3-adrenergic receptors and CEBPα (**k**) of rats after 9 days of the injection of Ad-GFP and Ad-DNp53RK in the ARC. β-actin and α-tubulin were used to normalize protein levels. Dividing lines indicate spliced bands from the same gel. Values are mean ± SEM of 6–8 animals per group. *$P < 0.05$; **$P < 0.01$

food intake (Fig. 3a, b). However, when fed a HFD, a significant increase in body weight, food intake, and adiposity was found among male AgRPp53 KO mice compared with wild type (WT) controls (Fig. 3a–d and Supplementary Fig. 5). AgRPp53 KO mice exhibited reduced locomotor activity and energy expenditure, without changes in respiratory quotient (Fig. 3e–h) or body length (Supplementary Fig. 5). In agreement with decreased energy expenditure, AgRPp53 KO mice fed a HFD had a lower interscapular surface temperature adjacent to BAT (Fig. 3i), more lipid droplets in the BAT (Fig. 3j), and downregulated protein levels of BAT UCP1 (Fig. 3k). In keeping to the increased adiposity, histomorphological analysis revealed significantly larger white adipocytes (Fig. 3l) and increased lipid content in the liver (Fig. 3m) in AgRPp53 KO mice following HFD exposure, but these mice did not exhibit differences in glucose tolerance or insulin sensitivity (Fig. 3n, o). Mice with genetic ablation of p53 in AgRP neurons showed a significant increase in food intake during the last weeks of HFD, much later than changes in body weight were detected, suggesting that the higher susceptibility to DIO occurs in a feeding-independent manner. Since AgRP neurons play a key role in the regulation of food intake, we next investigated whether p53 in AgRP neurons might be involved in the fasting-induced hyperphagia. Thus, mice were fasted overnight and then refed for 4 h. Mice lacking p53 in AgRP neurons

showed a non-significant trend toward decreased fasted-induced hyperphagia (Supplementary Fig. 6a). Alteration of circadian pattern of feeding has been associated with obesity[31] and p53 has been reported to interact with the circadian clock[32]. Therefore, we next assessed whether mice lacking p53 in AgRP fed a HFD for 10 weeks, when body weight was already increased compared to their littermates, but we failed to detect any change in food intake along the circadian cycle (Supplementary Fig. 6b, c).

Given the potentially pleotropic effects of p53 not only on cell activity but also on multiple biological actions, we measured AgRP fiber density in the ARC and paraventricular nucleus (PVH) of mice fed a HFD for 13 weeks and found that the number of fibers were identical between both genotypes (Supplementary Fig. 7). This indicates that the lack of p53 is not damaging these neurons at least not at this time point. In addition, we have also quantified the number of AgRP neurons and total number of cells in WT and conditional KOs and found that they were very similar between genotypes (Supplementary Fig. 8). Given the role of p53 in stress response and that hypothalamic inflammation influences energy balance[33], we performed immunostaining of GFAP and Iba1 in the hypothalamic ARC of mice and also measured protein levels of some inflammatory markers (IL-6 and IL-1β) in the mediobasal hypothalamus of WT and mice lacking p53 in AgRP neurons

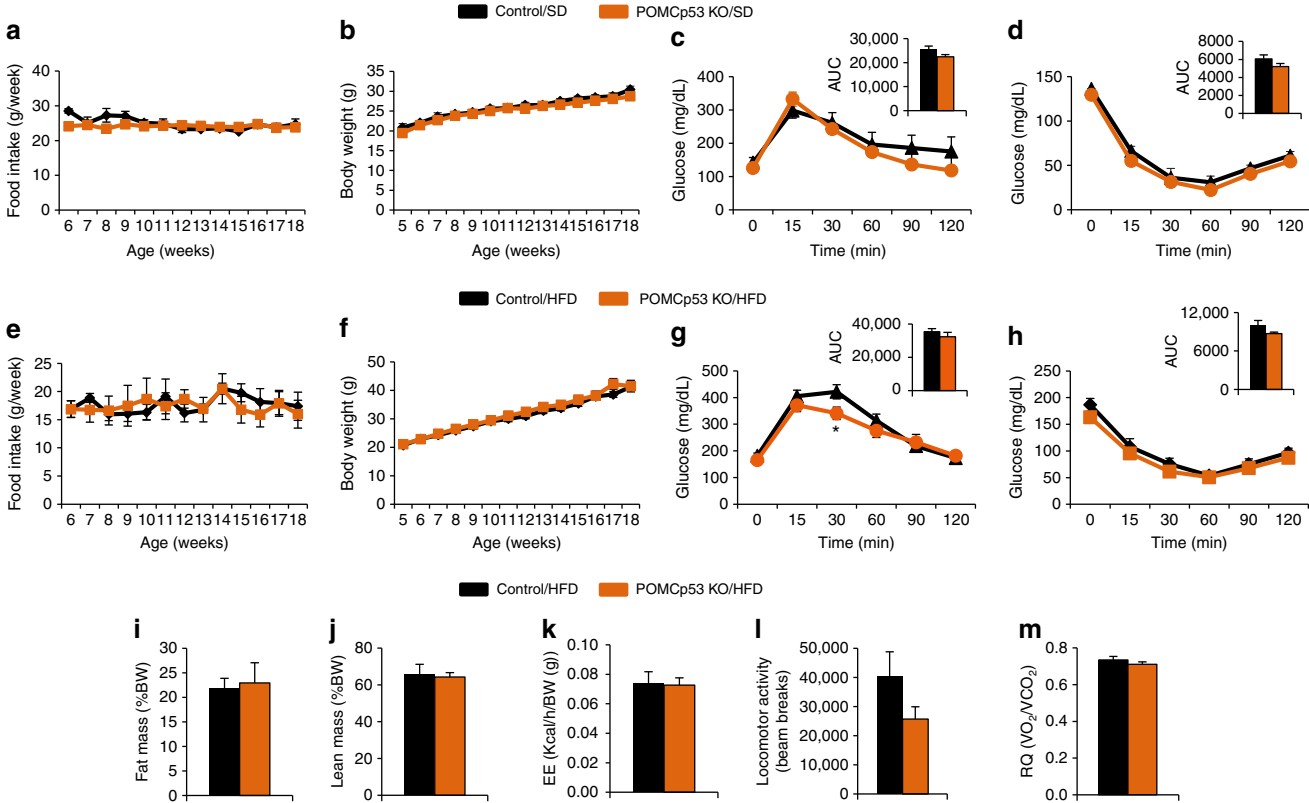

**Fig. 2** Ablation of p53 in POMC neurons does not alter energy balance. Food intake (**a**); body weight (**b**); glucose tolerance test (**c**); and insulin tolerance test (**d**) in control and POMCp53 KO mice fed a chow diet. Food intake (**e**); body weight (**f**); glucose tolerance test (**g**); insulin tolerance test (**h**); fat mass (**i**); lean mass (**j**); energy expenditure (EE) (**k**); locomotor activity (**l**); and respiratory quotient (RQ) (**m**) in control and POMCp53 KO mice fed a HFD. Values are mean ± SEM of 9–22 animals per group. *$P < 0.05$

after 13 weeks of HFD and no differences were found (Supplementary Fig. 9a–g). We also measured markers of apoptosis (caspase 3 and 7) and senescence (Poly (ADP-ribose) polymerase 1, PARP1) in control and AgRPp53 KO mice fed a HFD for 13 weeks. However, we did not find any change in these markers (Supplementary Fig. 9h).

**Ablation of p53 does not change AgRP neuron activity**. Next, we assessed the effect of AgRP neuron-specific ablation of p53 on the intrinsic electrophysiological properties of these neurons during DIO. To this end we performed perforated patch clamp recordings in brain slices of AgRP-Cre and AgRPp53 KO mice fed a HFD for 10 weeks and injected with AAV-DIO-hSyn-EGFP in the mediobasal hypothalamus. Both mouse lines showed a pattern of GFP label in the ARC, which is consistent with previously described AgRP expression pattern in the ARC. For post hoc identification, cells were loaded with biocytin by converting the perforated patch configuration to the whole-cell configuration at the end of the recording. The identity of AgRP neurons was then confirmed by double-labeling of the recorded neuron with GFP immunoreactivity and biocytin (Fig. 4). During the recording GABAergic and glutamatergic synaptic input was pharmacologically blocked to minimize influence from the network. Assessing several key electrophysiological parameters such as spontaneous firing rate, input resistance, spike frequency adaptation (SFA), spike threshold, and excitability did not reveal differences between the AgRP neurons in the two mouse lines (Fig. 4).

**UPR function reverses the phenotype of AgRPp53 ko mice**. Recent data demonstrate that hypothalamic activation of the

unfolded protein response (UPR) plays a primary pathogenic role in obesity. Diet-induced and genetic models of obesity have been associated with activated hypothalamic UPR[34–40]. We therefore evaluated the protein levels of UPR markers in our animal models. First, we found that rats injected with a mutant Tp53RK in the ARC showed diminished protein levels of BIP/GRP78, a chaperone that facilitates the proper protein folding acting upstream of the UPR[41], and increased protein levels of each UPR marker analyzed, Inositol requiring enzyme 1 (IRE1), activating transcription factor 6 alpha (ATF6α), and CCAAT/enhancer-binding protein-homologous protein (CHOP) (Fig. 5a). We also found increased protein levels of phosphorylated-SAPK/JNK (Thr183/Tyr185) (pJNK) and its upstream kinases[42], MKK4 and MKK7 in the mediobasal hypothalamus of AgRPp53 KO mice fed a HFD for 13 weeks (Fig. 5b). We next evaluated the protein levels of UPR markers in the mediobasal hypothalamus of AgRPp53 KO mice fed a HFD for 13 weeks, and similarly found decreased GRP78 and increased the levels of IRE1, pPERK, and ATF6α (Fig. 5b).

Our results indicate that the loss of p53 in the ARC is associated with increased ER stress, and we next aimed to determine the relevance of UPR as a mediator of the central actions of p53. For this purpose, we relieved ER stress using tauroursodeoxycholic acid (TUDCA), a chemical chaperone known to improve ER function and decrease the accretion of misfolded proteins in the ER lumen[43]. AgRPp53 KO and control littermates fed a HFD were treated intracerebroventricular (i.c.v.) with vehicle or TUDCA (5 µg/mouse/day) during 7 days. TUDCA administration did not modify food intake or body weight gain in WT mice (Fig. 5c, d). However, the central administration of TUDCA promoted a significantly anorectic and

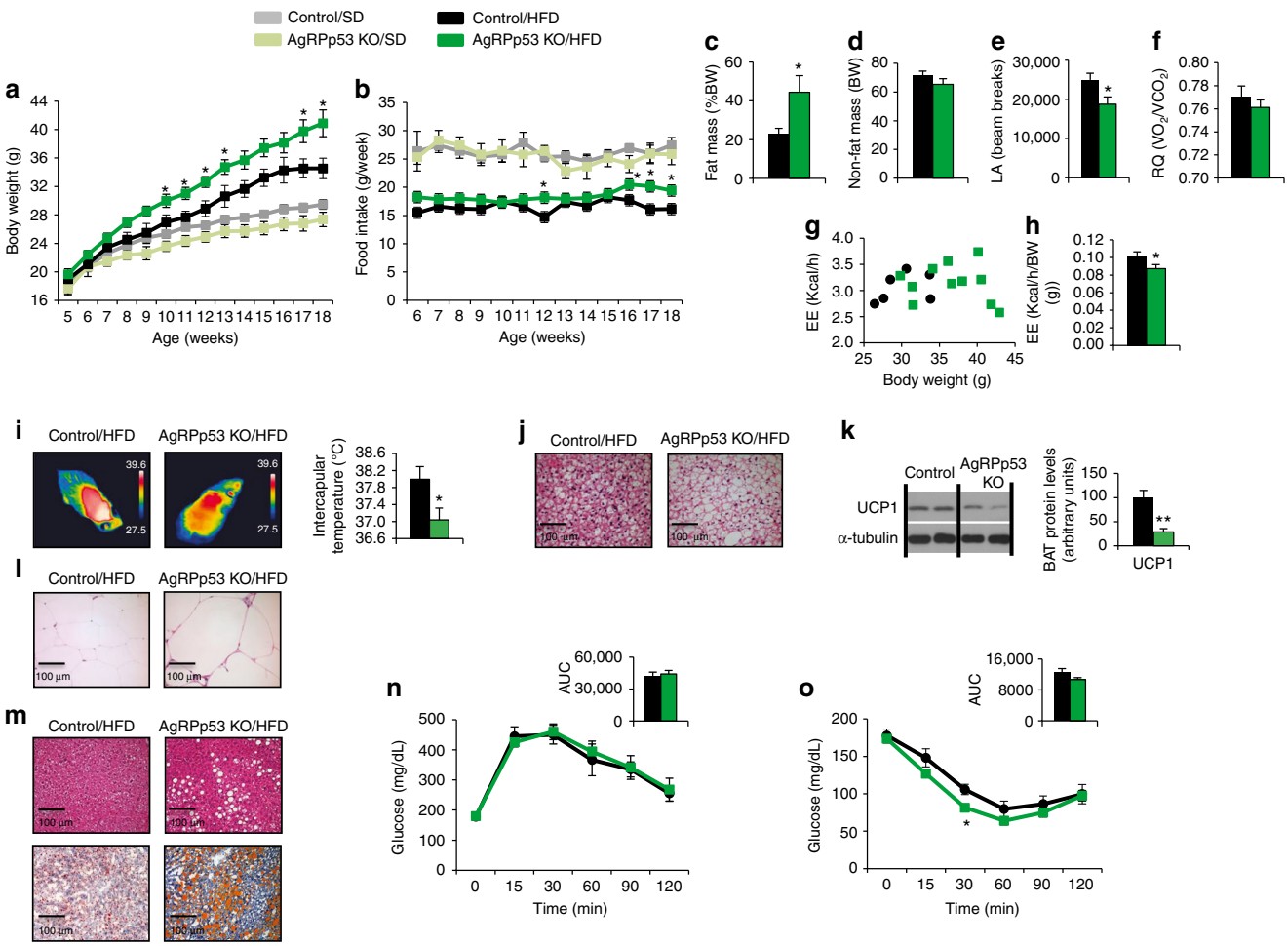

**Fig. 3** Ablation of p53 in AgRP neurons is associated with a positive energy balance during long-term HFD. Body weight (**a**) and food intake (**b**) of control and AgRPp53 KO mice fed a chow diet and HFD for 13 weeks. Fat mass (**c**); non-fat mass (**d**); locomotor activity (LA) (**e**); respiratory quotient (RQ) (**f**); energy expenditure (**g–h**); representative infrared thermal images and temperature of the BAT area (**i**); representative BAT histology pictures (hematoxylin-eosin) (**j**); BAT protein levels of UCP1 (**k**); representative WAT histology pictures (hematoxylin-eosin) (**l**); representative liver histology pictures (hematoxylin-eosin, upper and oil-red O staining, lower) (**m**); glucose tolerance test (**n**); and insulin tolerance test (**o**) of control and AgRPp53 KO mice fed a HFD. α-tubulin was used to normalize protein levels. Dividing lines indicate spliced bands from the same gel. Values are mean ± SEM of 5–15 animals per group. $*P < 0.05$; $**P < 0.01$

weight-reducing effect in AgRPp53 KO mice compared to vehicle-treated mice (Fig. 5c, e). Consistent with these results, adiposity was normalized in AgRPp53 KO mice treated with TUDCA, the size of white adipocytes and lipid droplets in BAT was reduced, and BAT UCP1 protein levels were elevated (Fig. 5f–i).

**JNK mediates the effects of AgRP p53 on energy balance**. Since we found that the inhibition of p53 activity in the ARC and the lack of p53 in AgRP neurons enhanced MKK4, MKK7, pJNK, and the levels of ER stress markers, we next assessed whether the upregulation of those factors could be observed before AgRPp53 KO mice develop obesity. We fed control and AgRPp53 KO mice a HFD for 3 weeks and then measured protein levels of MKK4, MKK7, pJNK, and ER stress markers in the mediobasal hypothalamus. At this time point there was a tendency of increased food intake, but there were no differences in body weight, energy expenditure, locomotor activity, respiratory quotient, lipid droplets in the BAT or BAT UCP1 levels between the two genotypes (Fig. 6a–i), whereas the protein levels of MKK7 and phosphorylated levels of JNK were already increased, while the levels of ER

stress markers remained unchanged (Fig. 6j). Thus, these results suggest that the upregulation of MKK7 and further activation of JNK precedes changes in hypothalamic ER stress and obesity in AgRPp53 KO mice fed a HFD.

Given these results, we hypothesized that JNK may play a key role in the development of obesity caused by the lack of p53 in AgRP neurons. To test this, AgRPp53 KO and control littermates fed a HFD were treated i.c.v. with vehicle or the JNK inhibitor SP-600125 (1.5 µg/mouse/day) during 6 days[44]. The administration of SP-600125 did not modify food intake or body weight gain in control mice (Fig. 7a–c). However, in AgRPp53 KO mice, central administration of SP-600125 decreased body weight and tended to inhibit food intake compared to vehicle (Fig. 7a–c). Consistently, AgRPp53 KO mice treated with SP-600125 showed reduced adiposity and white adipocyte size, while interscapular temperature was increased, BAT UCP-1 protein levels were stimulated and lipid droplets in BAT were reduced (Fig. 7d–i). Collectively, the data indicate that the central inhibition of JNK reverses the obese phenotype of AgRPp53 KO mice fed a HFD. Importantly, the dose of SP-600125 used did not modify body weight, food intake, or adiposity in WT mice fed a chow diet (Supplementary Fig. 10a–e). Although the i.c.v. administration of

SP-600125 increased protein levels of IL-1β and caspase 3 in the mediobasal hypothalamus (Supplementary Fig. 10f–g), the JNK inhibitor did not negatively affect AgRP fiber density in the ARC and PVH (Supplementary Fig. 11), indicating that this compound is not damaging these neurons.

To further analyze the role of JNK as a mediator of the effects of hypothalamic p53, we next inactivated hypothalamic p53 in WT and JNK1 KO mice fed a chow diet. The inhibition of p53 in the ARC of WT mice induced weight gain, food intake, and adiposity (Fig. 7j–m), reduced interscapular temperature (Fig. 7n),

increased lipid droplets in BAT (Fig. 7o), decreased BAT UCP1 protein levels (Fig. 7p, q), and increased the size of white adipocytes (Fig. 7r). However, all these effects were totally blunted in JNK1-deficient mice (Fig. 7j–r). Altogether, these results indicate that hypothalamic p53 requires JNK1 to exert its actions on energy homeostasis. Since the hypothalamic–pituitary–thyroid axis has been reported to be involved in the metabolic regulation by JNK[45], we measured serum concentration of T4 in WT and mice lacking p53 in AgRP neurons fed a HFD for 13 weeks. T4 levels were similar between both genotypes (Supplementary

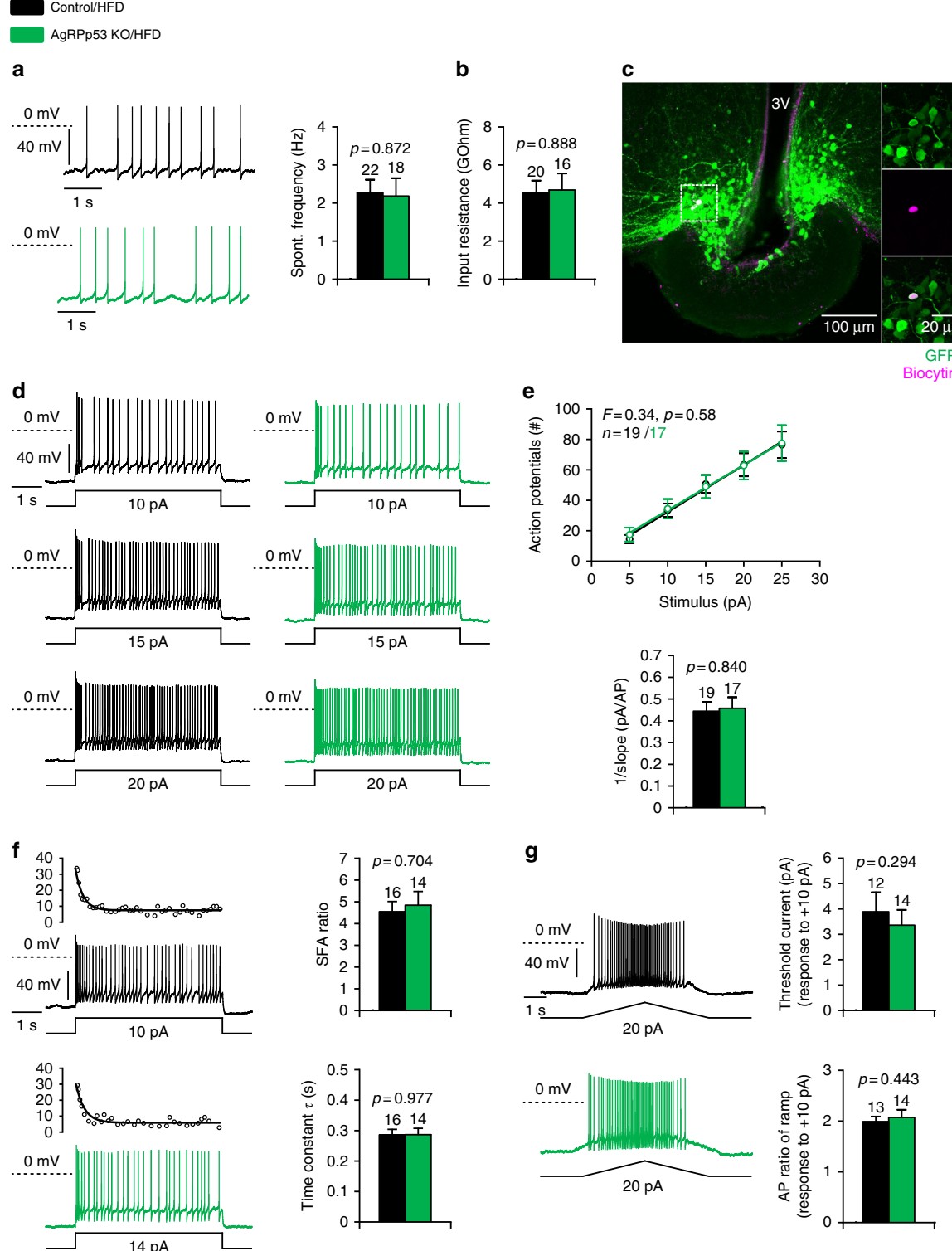

Fig. 12), suggesting that changes in thyroid status do not explain the phenotype of these mice.

**Activation of p53 in the ARC ameliorates DIO.** Since the inhibition of p53 in AgRP neurons induces weight gain and decreases BAT thermogenic activity, we hypothesized that the specific overexpression of p53 in the ARC of WT DIO mice might be sufficient to increase the BAT thermogenic activity and to ameliorate weight gain. For this, we injected adenoviruses encoding either Ad-GFP or Ad-p53+ together with GFP into the ARC. The efficacy of the stereotaxic injections in the ARC was corroborated by immunostaining of GFP (Fig. 8a) and the

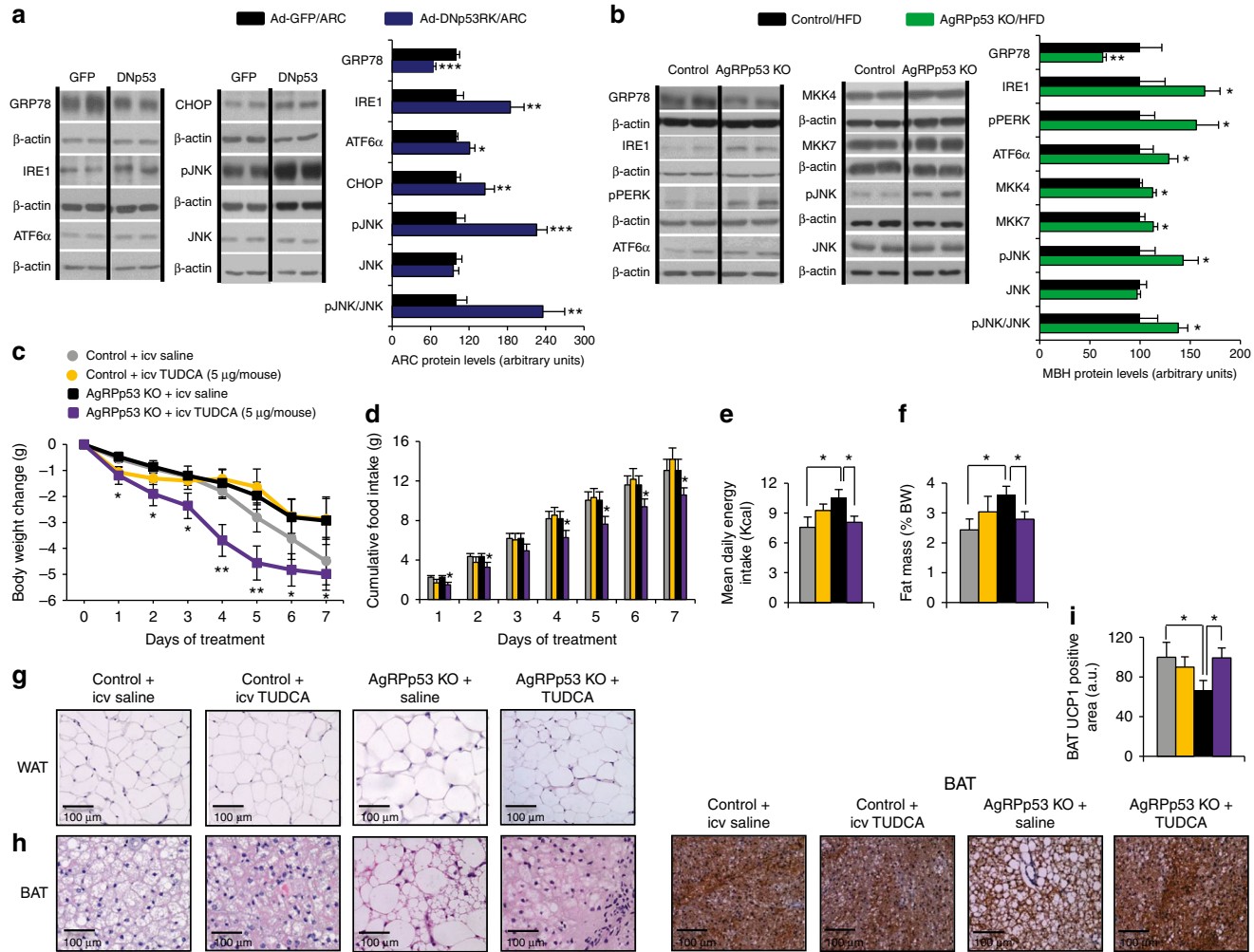

**Fig. 5** Improvement of ER function reverses the obese phenotype of AgRPp53 KO mice. Protein levels of GRP78, IRE1, ATF6α, CHOP, pJNK, and JNK in the ARC of rats after 9 days of the injection of Ad-GFP and Ad-DNp53RK (**a**). Protein levels of GRP78, IRE1, pPERK, ATF6α, MKK4, MKK7, pJNK, and JNK in the mediobasal hypothalamus of control and AgRPp53 KO mice fed a HFD for 13 weeks (**b**). Effects of 7 days i.c.v. injections of TUDCA (5 μg/mouse) or vehicle to control and AgRPp53 KO mice fed a HFD on body weight change (**c**); cumulative food intake (**d**); mean daily energy intake (**e**); adiposity (**f**); representative WAT histology pictures (hematoxylin-eosin) (**g**); representative BAT histology pictures (hematoxylin-eosin) (**h**); and immunostaining of UCP1 in BAT (**i**). β-actin was used to normalize protein levels. Dividing lines indicate spliced bands from the same gel. Values are mean ± SEM of 5–10 animals per group. *$P < 0.05$; **$P < 0.01$; ***$P < 0.001$

**Fig. 4** Cell type-specific ablation of p53 does not change AgRP neurons' electrophysiological properties in DIO. Electrophysiological parameters of AgRP neurons that were recorded in AgRP-Cre mice (black) and AgRPp53 KO mice (green) and a HFD for 10 weeks and injected in the mediobasal hypothalamus with AAV-DIO-hSyn-EGFP that expressed GFP specifically in AgRP neurons. **a** Spontaneous activity. Original recordings (left) and mean frequencies (right). **b** Mean input resistance. **c** Immunohistochemical identification of a recorded AgRP neuron expressing GFP (green) that was loaded with biocytin (red) during the electrophysiological recording in the ARC. Excitabilty. **d** Representative voltage responses to increasing depolarizing current injections (from 10 to 20 pA in 5 pA steps) elicited from a holding potential of −70 mV. **e** Top: Mean spike count during the current pulses as a function of injected current. Bottom: Mean increase in firing in response to the injected current (pA$^{-1}$). **f** Spike frequency adaptation. Left: Original recording and corresponding plots of the instantaneous spike frequency. Right: Mean SFA ratio and mean time constant of decay in instantaneous frequency. Analysis was performed from traces with similar instantaneous frequency (30–35 Hz). **g** Current ramps were used to measure the spike threshold and voltage-dependent sustained changes in activity. Left: Original recording. Right: (top) Mean threshold current to elicit action potentials; (bottom) ratio of spike count between on and off ramp. Data are represented as mean ± SEM. N-values and p-values (unpaired Student's t-test) are given above the bar graphs

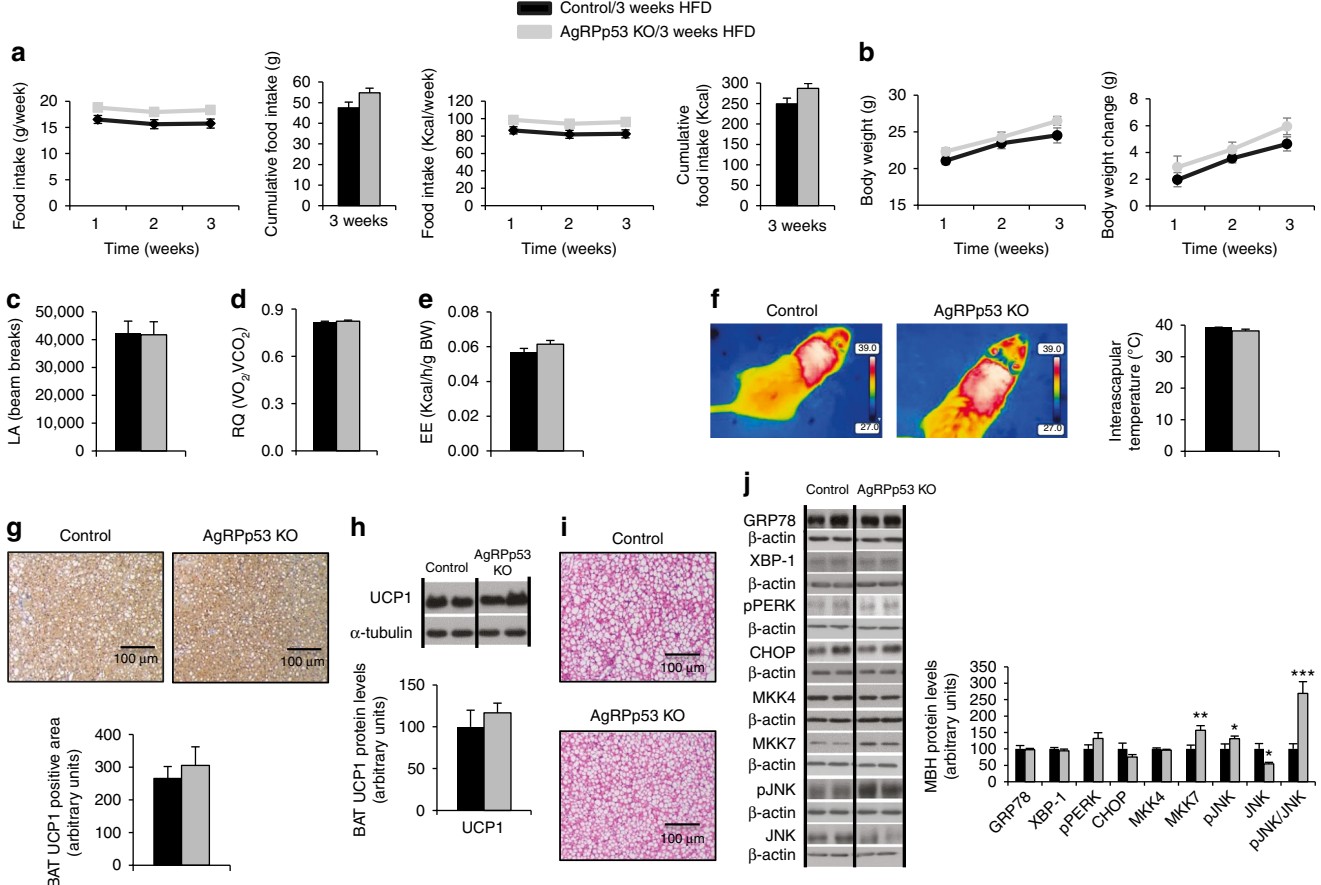

**Fig. 6** Hypothalamic protein levels of MKK7 and pJNK are increased in AgRPp53 KO mice fed a HFD before displaying obesity. Food intake (**a**) and body weight (**b**) of control and AgRPp53 KO mice fed a HFD for 3 weeks. Locomotor activity (LA) (**c**); respiratory quotient (RQ) (**d**); energy expenditure (**e**); representative infrared thermal images and temperature of the BAT area (**f**); immunostaining and protein levels of UCP1 in BAT (**g–h**); representative BAT histology pictures (hematoxylin-eosin) (**i**); protein levels of GRP78, XBP1, pPERK, CHOP, MKK4, MKK7, pJNK, and JNK in the mediobasal hypothalamus of control and AgRPp53 KO mice fed a HFD for 3 weeks (**j**). β-actin and α-tubulin were used to normalize protein levels. Dividing lines indicate spliced bands from the same gel. Values are mean ± SEM of 4–9 animals per group. *$P < 0.05$; **$P < 0.01$; ***$P < 0.001$

functional efficiency of the viral vectors was demonstrated by increased protein levels of phosphorylated p53 in the ARC (Fig. 8b). The p53 overexpression in the ARC of DIO mice significantly decreased body weight and food intake (Fig. 8c, d). This was associated with decreased adiposity without changes in non-fat mass, increased energy expenditure, unchanged locomotor activity, and decreased respiratory quotient (Fig. 8e–i). The overexpression of p53 in the ARC of DIO mice increased interscapular surface temperature in the BAT area (Fig. 8j), reduced the lipid droplet size in BAT (Fig. 8k), and stimulated the expression of BAT UCP1 (Fig. 8l). Accordingly, the size of white adipocytes and the hepatic lipid content was reduced (Fig. 8m, n). At the hypothalamic level, p53 overexpression in the ARC attenuated DIO-induced ER stress, as indicated by the lower protein levels of IRE, pPERK, CHOP, as well as MKK4, MKK7, and pJNK (Fig. 8o). The overexpression of p53 also decreased protein levels of caspase 7, without affecting caspase 3 or PARP in the mediobasal hypothalamus (Supplementary Fig. 13). To investigate the effects of the overexpression of p53 when diet is modulated, we overexpressed p53 in mice fed a chow diet. Under these conditions, we did not find any change in body weight or other metabolic parameters measured like food intake, body composition, BAT thermogenesis, or adipocyte size (Supplementary Fig. 14). These new results reinforce the hypothesis that the

effects of hypothalamic p53 on body weight and metabolism occur only in a HFD setting.

To further explore the role of p53 in AgRP neurons, we next performed an experiment where we recovered the expression of p53 specifically in AgRP neurons of AgRPp53 KO mice fed a HFD injecting AAV-hSyn-DIO-p53. As expected, AgRPp53 KO mice gained more weight than their controls when put under a HFD (mice were fed a HFD for 10 weeks before the viral injections), but when these mice were injected with the AAV-hSyn-DIO-p53, they lost weight after 4 weeks in a food intake-independent manner (Fig. 9a–c), which was consistent with a reduction in fat mass, decreased adipocyte size, increase in BAT temperature, and higher BAT UCP1 immunostaining (Fig. 9d–i). Therefore, these results suggest that p53 specifically overexpressed in AgRP neurons is sufficient to produce a catabolic action in obese mice.

In addition to those genetic approaches, we assessed whether central pharmacological stimulation of p53 would mimic the findings obtained after genetic overexpression. First, we treated rats with i.c.v. adriamycin (a drug activating p53) using osmotic minipumps delivering 1 μg/rat/day for 5 days and found a significant decrease in food intake and weight gain (Supplementary Fig. 15a, b). The control of BAT thermogenesis is regulated by the SNS via the β-adrenergic receptors. To test if the effects of

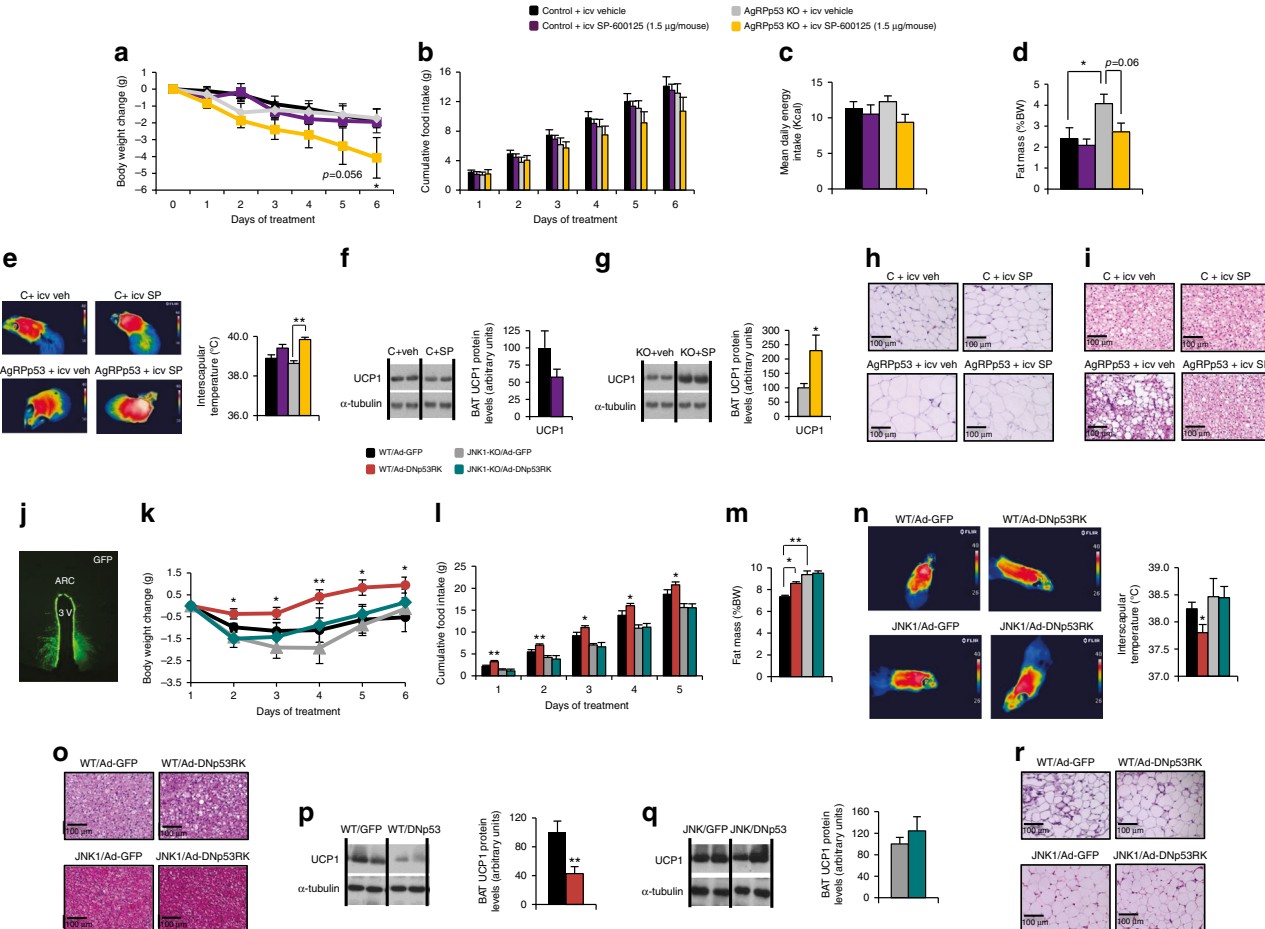

**Fig. 7** Inhibition of central JNK reverses the obese phenotype of AgRPp53 KO mice. Body weight change (**a**); cumulative food intake (**b**); mean daily energy intake (**c**); fat mass (**d**); representative infrared thermal images and temperature of the BAT area (**e**); BAT protein levels of UCP1 in control mice (**f**) and AgRPp53 KO mice (**g**); representative WAT histology pictures (hematoxylin-eosin) (**h**); and representative BAT histology pictures (hematoxylin-eosin) (**i**) after i.c.v. injections of SP-600125 (1.5 μg/mouse) or vehicle in control and AgRPp53 KO mice fed a HFD. Representative immunofluorescence showing GFP expression in the ARC (**j**); body weight change (**k**); cumulative food intake (**l**); fat mass (**m**); representative infrared thermal images and temperature of the BAT (**n**); representative BAT histology pictures (hematoxylin-eosin) (**o**); BAT protein levels of UCP1 in control mice (**p**) and JNK1 KO mice (**q**) and representative WAT histology pictures (hematoxylin-eosin) (**r**) of control and JNK1 KO mice after 6 days of the injection of Ad-GFP and Ad-DNp53RK in the ARC. α-tubulin was used to normalize protein levels. Dividing lines indicate spliced bands from the same gel. Values are mean ±4–10 animals per group. *$P < 0.05$; **$P < 0.01$

hypothalamic p53 on BAT thermogenesis were driven by the SNS, we treated rats with a β3 receptor antagonist (SR59230A) together with i.c.v. adriamycin. We found that the reduction in weight gain in animals treated with i.c.v. adriamycin was blocked when the selective β3 receptor antagonist was subcutaneously co-administered (Supplementary Fig. 15b) at a dose that does not affect body weight per se[46]. Since the inhibition of p53 in the ARC altered energy homeostasis, we next performed an experiment infusing adriamycin directly in the ARC of rats using osmotic minipumps (Supplementary Fig. 15c). Body weight and fat mass were reduced independently of changes in food intake or lean mass (Supplementary Fig. 15d–g), protein levels of pJNK in the ARC were downregulated, BAT lipid droplets were reduced, and BAT UCP1 protein levels were stimulated (Supplementary Fig. 15h–j) in ARC adriamycin-treated animals in comparison to vehicle rats.

**p53 in AgRP neurons is essential for ghrelin**. Given the critical role played by p53 in AgRP neurons, we decided to investigate whether it could also be involved in the signaling pathway

mediating the central effect of ghrelin[47,48]. We have previously shown that central p53 plays an important role mediating ghrelin-induced food intake[28]. Taking into account that the main neuronal population targeted by ghrelin is AgRP[49,50], we tested the ability of central ghrelin to induce food intake in AgRPp53 KO mice. A single i.c.v. injection of ghrelin induced food intake and body weight in control mice but not in AgRPp53 KO (Fig. 10a, b). In agreement with this, AgRP and NPY mRNA expression was increased in control mice, while this effect was not found in AgRPp53 KO mice (Fig. 10c). The larger size of white adipocytes found in control mice treated with ghrelin was not detected in AgRPp53 KO-treated mice (Fig. 10d). Finally, the ghrelin-induced expression of fatty acid synthase (FAS), acetyl CoA carboxylase (ACC), and lipoprotein lipase (LPL) in control mice (Fig. 10e) was not detected in AgRPp53 KO mice treated with ghrelin (Fig. 10f). Finally, we also found higher levels of acyl-ghrelin in AgRPp53 KO compared to control mice (Fig. 10g), suggesting an overcompensating ghrelin resistance. Collectively, our data indicate that p53 located in AgRP neurons is a key modulator of ghrelin-induced feeding and adiposity.

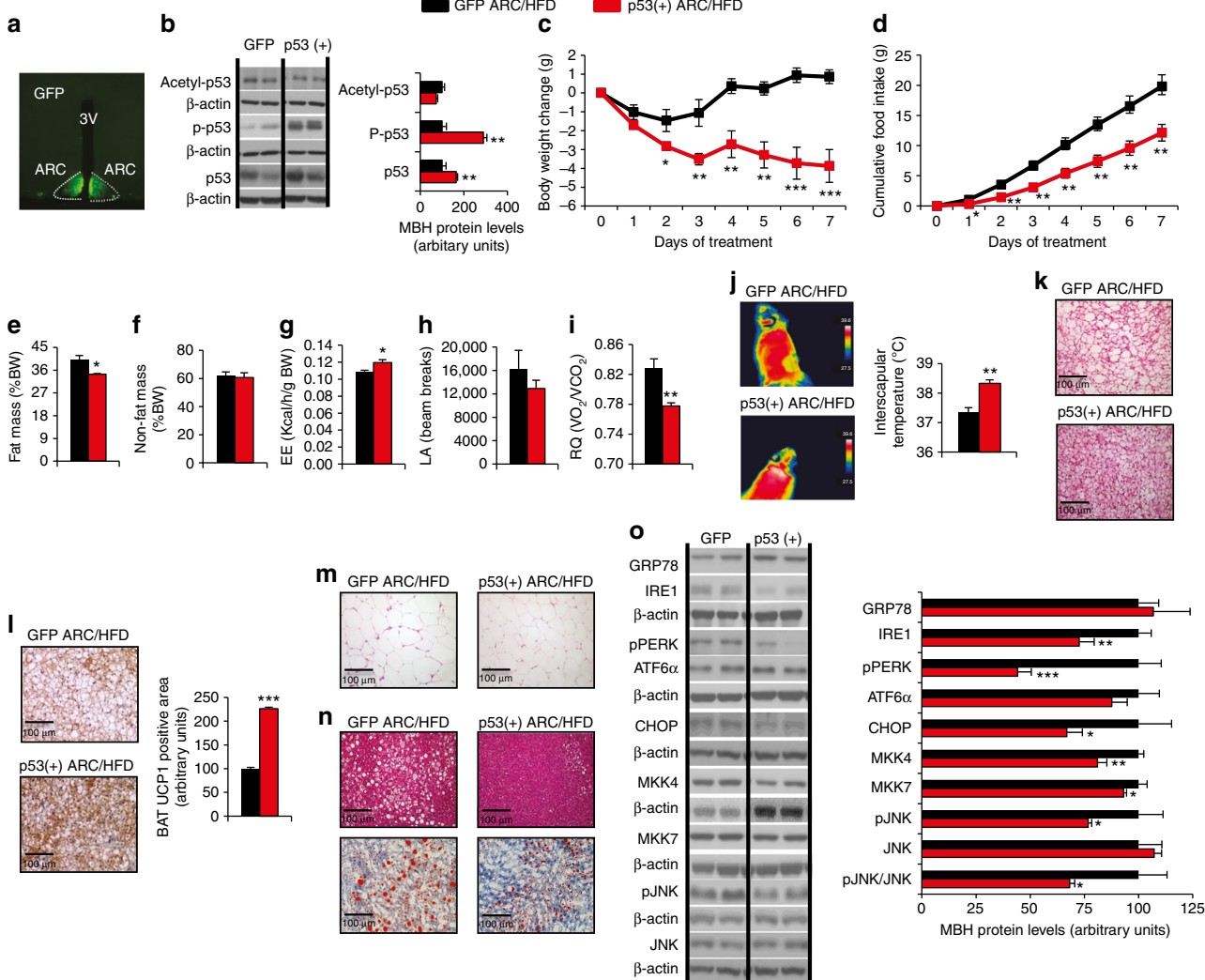

**Fig. 8** Genetic activation of p53 in the ARC ameliorates diet-induced obesity. Representative immunofluorescence images showing GFP expression in the ARC (**a**); protein levels of acetyl-p53, phosphorylated p53, and in the mediobasal hypothalamus (**b**); body weight change (**c**); cumulative food intake (**d**); fat mass and non-fat mass (**e–f**); energy expenditure (EE) (**g**); locomotor activity (LA) (**h**); respiratory quotient (RQ) (**i**); representative infrared thermal images and temperature of the BAT area (**j**); representative BAT histology pictures (hematoxylin-eosin) (**k**); immunostaining of UCP1 in BAT (**l**); representative WAT histology pictures (hematoxylin-eosin) (**m**); representative liver histology pictures (hematoxylin-eosin, upper and oil-red O staining, lower) (**n**); and protein levels of GRP78, IRE1, pPERK, ATF6α, CHOP, MKK4, MKK7, pJNK, and JNK in the mediobasal hypothalamus (**o**) of DIO mice after 7 days of the injection of Ad-GFP and Ad-p53 in the ARC. β-actin was used to normalize protein levels. Dividing lines indicate spliced bands from the same gel. Values are mean ± SEM of 4–8 animals per group. *$P < 0.05$; **$P < 0.01$; ***$P < 0.001$

## Discussion

The interest in p53 has classically been focused in its anti-tumorigenic action, but has expanded into the field of metabolism by the findings that cancer cells must maintain an accelerated metabolic rate to cope with the energetic requirements of high replication. In keeping with this, over the last few years, a large set of data indicates that p53 plays an important metabolic role[12,15], with some of the evidence as follows: p53 has been shown to modulate the activity of several key metabolic tissues, including the WAT, where it is induced in obese, leptin-deficient mice[21], and where p53 upregulation led to insulin resistance[20]. p53 is also present in liver, where it modulates lipid metabolism and is increased in mouse models of fatty liver disease[29]. In BAT, the role of p53 is still unclear but it is suggested to mediate BAT thermogenic activity[23,24,51]. In contrast to the insight of p53 action in peripheral tissues, the metabolic role of p53 in the hypothalamus remains almost completely unknown.

Consistent with a key role for p53 in metabolic regulation, our pharmacological and genetic approaches here establish hypothalamic p53 as a relevant player in the maintenance of energy homeostasis. Deletion of p53 in AgRP neurons caused a dramatic obese phenotype characterized by increased adiposity, slight hyperphagia, and reduced energy expenditure in a setting of nutrient excess. Importantly, these p53-related alterations in energy balance were found to be nuclei-specific, since they were only observed when p53 was manipulated in AgRP neurons and not when it was ablated in the other main neuronal population of the ARC, namely, POMC neurons or in SF1 neurons in the VMH. The phenotype obtained following deletion of p53 in AgRP neurons was difficult to predict since acute genetic ablation of AgRP in adult animals leads to anorexia, whereas its neonatal depletion reduces food intake and alters inter-organ communication, redirecting peripheral nutrient utilization toward increased fat oxidation[52,53]. This translates into obesity and

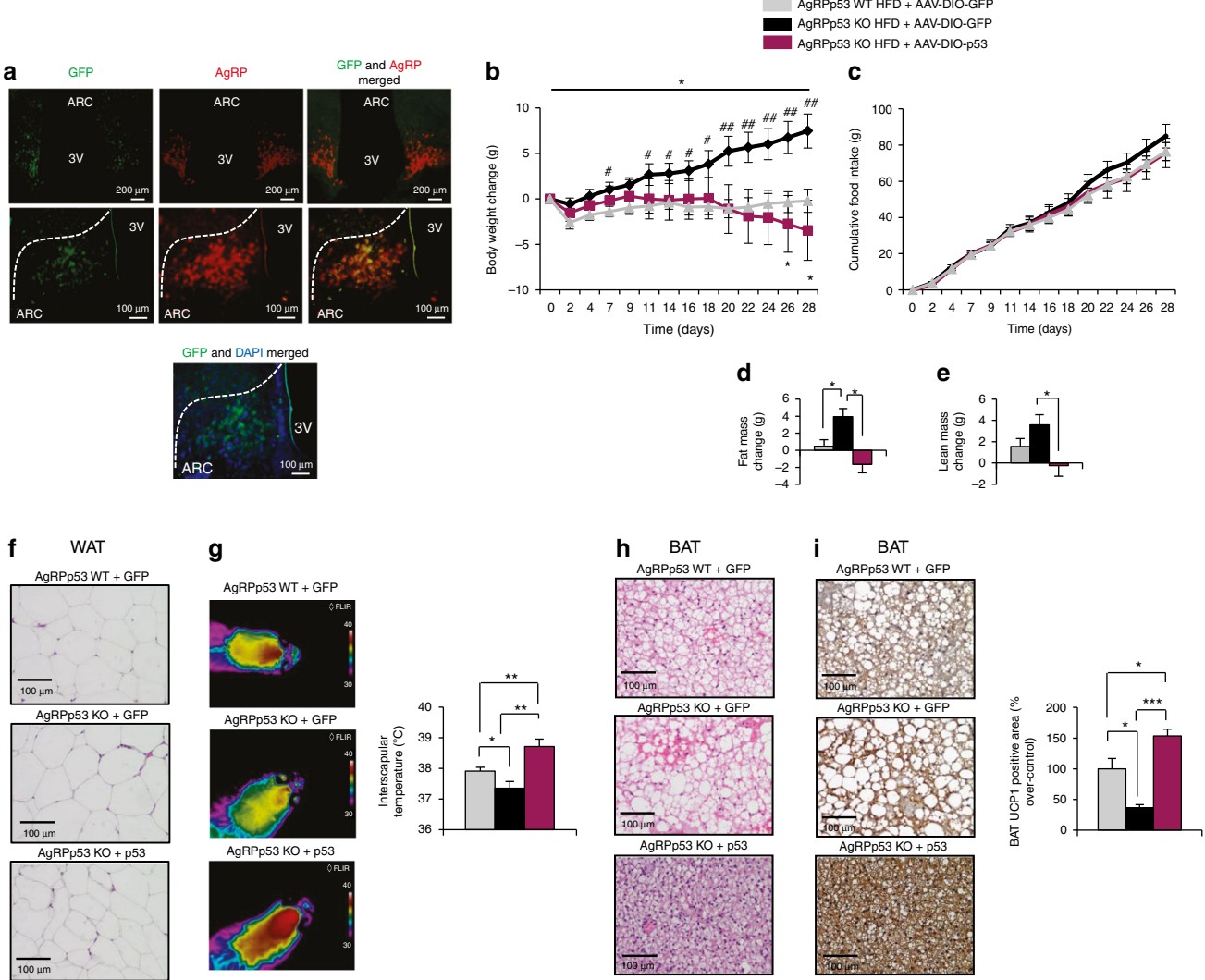

**Fig. 9** Genetic recovery of p53 in AgRP neurons decreases body weight gain. Representative immunofluorescence images showing the colocalization of GFP (green) and AgRP (red) in the ARC, as well as the colocalization of GFP (green) and DAPI (blue) (**a**); body weight change (**b**); cumulative food intake (**c**); fat mass and non-fat mass (**d–e**); representative WAT histology pictures (hematoxylin-eosin) (**f**); representative infrared thermal images and temperature of the BAT area (**g**); representative BAT histology pictures (hematoxylin-eosin) (**h**); and BAT UCP1 immunostaining (**i**) of AgRPp53 KO mice injected with AAV-DIO-hSyn-EGFP or AAV-DIO-hSyn-EGFP-p53 in the ARC. Values are mean ± SEM of 5–8 animals per group. *$P < 0.05$; **$P < 0.01$; ***$P < 0.001$

hyperinsulinemia on regular chow, but reduced body weight gain and paradoxical improves glucose tolerance on HFD[54]. In this regard, our AgRPp53 KO mice show a similar phenotype, which indicates that p53 plays a key role in some, but not all, the functions of AgRP neurons, something that is consistent with the fact that p53 is not expressed in 100% of AgRP neurons. Interestingly, genetic ablation of p53 in AgRP neurons did not change the electrical activity of these neurons. While we performed an unbiased sampling of the GFP-labeled AgRP neurons, the reason for this might be the heterogeneity of AgRP neurons and the existence of different neuronal subsets. For instance, there is a considerable differential gene expression among AgRP neurons after food deprivation[55]. Alternatively, FACS sorting and single-cell RNA sequencing showed that 69% of AgRP neurons express p53. Therefore, it might be possible that some of the recorded neurons would not be affected in our animal model, masking potential effects of p53 on AgRP electrical activity.

Different intracellular processes within AgRP neurons have been demonstrated to control energy homeostasis, including autophagy[56], mitochondrial dynamics[57], or inflammation[36]. Hypothalamic ER stress has also emerged as a relevant factor in the development of obesity[34–39,58]. In the current study, we found that genetic inhibition of p53 activity in the ARC, and more specifically its deletion in AgRP neurons, increased the hypothalamic expression of ER stress markers, while overexpression of p53 in the ARC attenuated DIO-induced hypothalamic ER stress. In this regard, the recovery of p53 in AgRP neurons of AgRPp53 KO mice fed a HFD, reversed their obese phenotype, and stimulated BAT thermogenesis. Moreover, our results are also in agreement with an in vitro study demonstrating that loss of p53 function activates the IRE1α/XBP1 pathway to enhance protein folding, suggesting that defects in the p53 complex represents a new mechanism for ER function modulation[59,60]. Furthermore, we found that obese AgRPp53 KO mice centrally treated with TUDCA showed a reduction in food intake, body weight, and adiposity. These results are consistent with previous reports indicating that reversal of central ER stress with chemical chaperones is able to decrease food intake and adiposity[34,36,38,43].

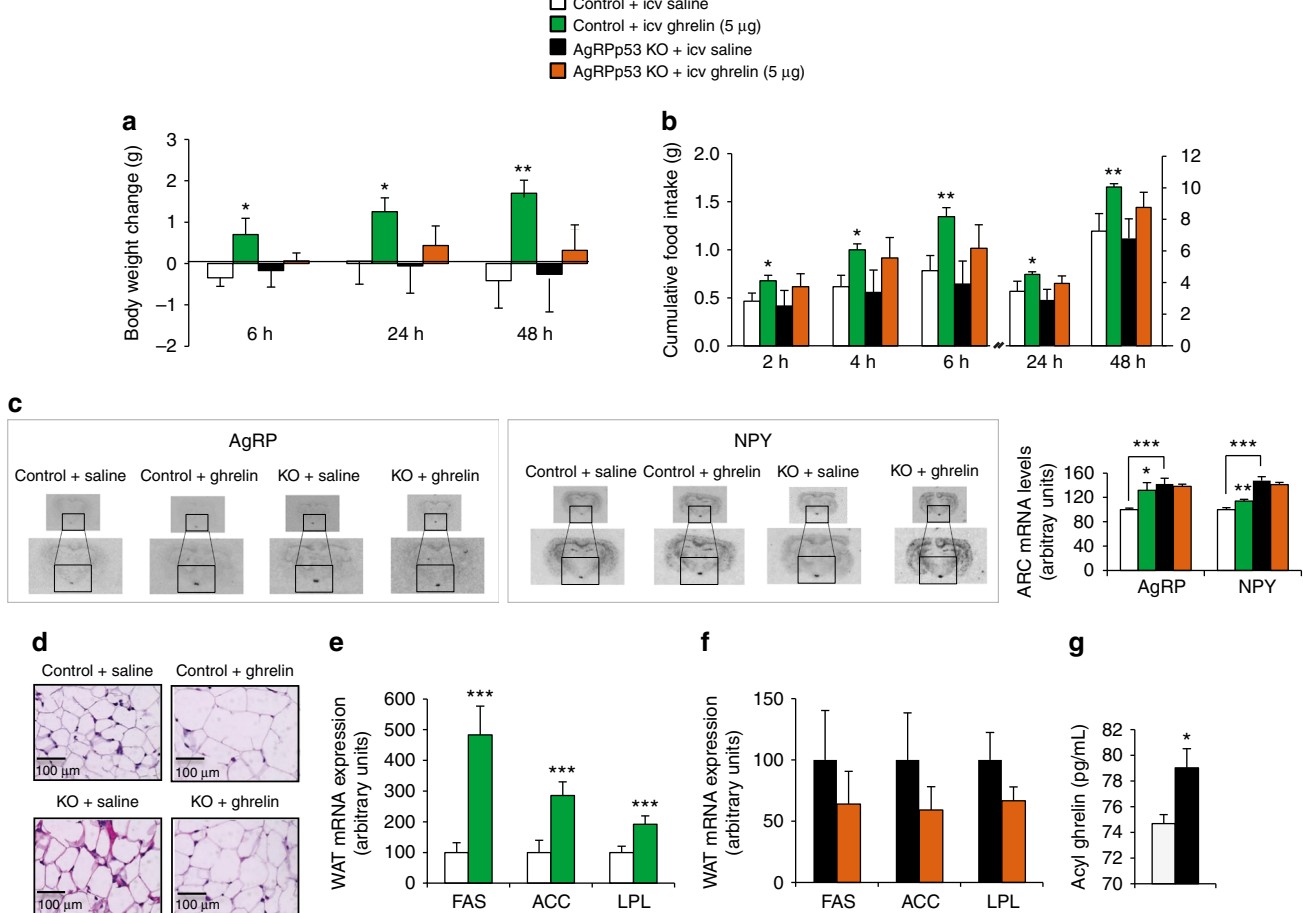

**Fig. 10** Central ghrelin fails to stimulate food intake and body weight in AgRPp53 KO mice. Body weight change (**a**); cumulative food intake (**b**); ARC mRNA expression of NPY and AgRP (**c**); representative WAT histology pictures (hematoxylin-eosin) (**d**); mRNA expression of FAS, ACC, and LPL in the WAT of control (**e**) and AgRPp53 KO mice (**f**) and serum acyl-ghrelin levels in control and AgRPp53 KO mice treated i.c.v. with saline or ghrelin. Values are mean ± SEM of 6–9 animals per group. *$P < 0.05$; **$P < 0.01$

Despite the fact that lack of p53 in AgRP neurons activates hypothalamic ER stress, this effect does not seem to be the primary cause leading to energy imbalance, as 3 weeks of HFD in AgRPp53 KO did not cause differences in body weight or expression of ER stress markers in comparison to their control littermates. This indicates that hypothalamic ER stress is likely a consequence rather than the cause of the obese phenotype. In contrast, we found that phosphorylated levels of JNK in the hypothalamus were increased in AgRPp53 KO mice before they developed obesity. Protein levels of MKK7, one of the upstream kinases responsible of JNK phosphorylation, also displayed significant elevation in the mediobasal hypothalamus of AgRPp53 KO mice before body weight was increased. JNK in AgRP neurons is known to play an important role in the control of energy balance; for instance, JNK3 deficiency was associated with enhanced excitatory signaling by AgRP neurons in HFD-fed obese mice[61], and in line with our results genetic activation of JNK1 signaling in AgRP neurons is sufficient to induce weight gain and adiposity in mice[62]. Consistently, the pharmacological inhibition of central JNK reversed the obese phenotype of AgRPp53 KO mice, by decreasing body weight and increasing BAT thermogenic activity. Moreover, p53 inhibition caused energy imbalance in WT but not in JNK1-deficient mice. Although the regulation of JNK activity by p53 has been studied largely in the context of apoptosis and autophagy[63,64], our findings indicate that the hypothalamic p53/JNK pathway is also

relevant in the control of energy balance. In this regard, it is important to note that obese AgRPp53 KO mice did not show changes in markers of inflammation, apoptosis or senescence in the hypothalamus, suggesting that the higher susceptibility to DIO is independent of those factors. Indeed, we cannot discard that hypothalamic p53 might have a role in apoptosis and senescence at higher ages, since our experiments were performed in relatively young mice that did not present any sign of tumor.

Finally, since the ARC is the primary hypothalamic area reached by circulating signals, we assessed the possibility that p53 in AgRP neurons might also act as a mediator of ghrelin actions. In this sense, we have previously shown that p53 is essential for the orexigenic and adipogenic action of ghrelin[28,65]. Taking into account that AgRP neurons are the main target for ghrelin[49,50,66], we found that, contrary to control mice, ghrelin was unable to induce food intake and body weight gain in AgRPp53 KO mice. Thus, p53 specifically located in AgRP neurons is essential for the orexigenic and adipogenic actions of ghrelin. Moreover, acyl-ghrelin levels were significantly increased in AgRPp53 KO3 mice, suggesting that these mice are overcompensating ghrelin resistance by increasing the circulating levels of active ghrelin.

Overall, our findings point to a fundamental and specific role for p53 in AgRP neurons in the regulation of energy homeostasis. The high susceptibility to obesity of AgRPp53 KO mice is correlated with a decreased energy expenditure and BAT thermogenic activity. Importantly, these metabolic alterations are

mediated by hypothalamic JNK activity. Finally, the genetic overexpression of hypothalamic p53 in the ARC or specifically in AgRP neurons of obese mice decreased body weight and increased BAT activity. These results support that beyond its actions as a tumor suppressor, hypothalamic p53 is critical to maintain energy balance.

## Methods

**Animal care.** 8–10-week-old (250–300 g) male Sprague Dawley rats and C57/BL6 WT mice were housed under conditions of controlled temperature (23 °C) and illumination (12-h light/12-h dark cycle). They were allowed ad libitum access to water and standard chow (Harlan Research Laboratories) or HFD (60% Kcal fat; Research Diets) (starting at 5 weeks of age). Care of all animals was within institutional animal care committee guidelines, and all procedures were reviewed and approved by the Ethics Committee of the University of Santiago de Compostela, in accordance with European Union normative for the use of experimental animals.

**Generation of p53-conditional knock out mice.** We used the cre-lox system to delete p53 in AgRP, POMC, or SF-1 neurons. p53$^{loxP/loxP}$ mice were purchased from The Jackson Laboratory (B6; 129S4 Trp53$^{tm5Tyj}$/J). To investigate the role of AgRP, POMC, or SF-1-expressing neurons, we inactivated the p53 gene specifically in each of these neurons by crossing p53$^{loxP/loxP}$ mice with mice expressing Cre recombinase under control of the AgRP, POMC, or SF-1 promoter[67]. Cre-mediated recombination was visualized by the use of a reporter mouse strain with the Cre inducible tdTomato-reporter mouse B6.Cg-Gt(ROSA)26Sor$^{tm14(CAG-tdTomato)Hze}$/J (Jackson Laboratories)[30].

**Physiological measurements.** Body weights and food intake were determined weekly. Whole-body composition was measured using NMR imaging (Whole Body Composition Analyzer; EchoMRI, Houston, TX). Animals were monitored in a custom 12-cage indirect calorimetry, food intake, and locomotor activity monitoring system (TSE LabMaster, TSE Systems, Germany) as previously described[68,69]. Mice were acclimated for 48 h to the test chambers and then were monitored for an additional 48 h. Data collected from the last 48 h was used to calculate all parameters for which results are reported. Heat production was visualized using a high-resolution infrared camera (FLIRSystems) as previously described[68]. Plasma Glucose tolerance tests and insulin sensitivity tests were performed on 4–6 h food-deprived mice. D-glucose (2 g/kg) and insulin (0.7 IU/kg) were injected intraperitoneally (i.p.) and blood glucose determined at the indicated time points. Blood glucose was measured using a Glucometer (Arkray) from the tail vein. Circulating T4 levels were measured by Total Thyroxine (T4) ELISA kit from Crystal Chem (Ref. 80983) according to manufacturer instructions. Animals were killed by decapitation and were not food-deprived before sampling. Tissues were removed rapidly, frozen immediately on dry ice, and kept at −80 °C until analysis.

**Stereotaxic microinjection of adenoviral expression vectors.** Ad libitum fed rats were anesthetized by an i.p. injection of ketamine (100 mg/kg body weight)/ xylazine (15 mg/kg body weight). Adenoviral vectors encoding a double-mutant kinase dead p53 regulating kinase (DNTp53RK) acting as a dominant negative (the TP53RK clone was then mutated at g217a and a218t to E73M) that block p53 activation and function (Ad-DNp53RK) (1.0 × 10^10 PFU/mL) (SL115889; Signa-Gen Laboratories, Gaithersburg, MD) or GFP controls (1.0 × 10^10 PFU/mL) (SL100708) were injected bilaterally into the ARC AP: −2.85 mm, L: ±0.3 mm, and V: −10.2 mm, with a 25-gauge needle (Hamilton) or into VMH AP: −2.4/3.2 mm, L: ±0.6 mm, and V: −10.1 mm, as previously reported[70–72].

C57BL/6 mice were fed with HFD for 12 consecutive weeks (starting at 5 weeks of age). Mice were anesthetized by an i.p. injection of ketamine/xylazine cocktail (ketamine 15 mg/kg BW/xylazine 3 mg/kg BW) and placed in a stereotaxic frame (Kopf Instruments). Adenoviral vectors containing the tumor protein p53 (1.0 × 10^10 PFU/mL) (SL109638) or Ad-GFP (1.0 × 10^10 PFU/mL) (SL100708; Signagen Laboratories) were injected in the ARC. The ARC was targeted bilaterally using a 32-gauge needle connected to a 1-ml syringe (Neuro-Syringe, Hamilton) and adenoviral vectors were delivered at a rate of 0.1 μl/min for 7 min (0.7 μl/injection site) according to the following coordinates: −1.5 mm posterior to the bregma, ±0.2 mm lateral to midline, and −6 mm below the surface of the skull as previously reported.

Ad libitum fed C57BL/6 mice or JNK knock out mice were anesthetized and placed in a stereotaxic frame (Kopf Instruments). Ad-DNp53RK or GFP controls were injected bilaterally into the ARC. Correct ARC adenoviral delivery was assessed by immunofluorescence analysis. After the procedure, the incision was closed with sutures, and mice were placed in a heated cage until they recovered from anesthesia. Body weight and food intake were measured daily for 5 consecutive days after the surgery.

To rescue the expression of p53 specifically in AgRP neurons, we injected in the ARC AAV-DIO-hSyn-EGFP or AAV-DIO-hSyn-p53-EGFP (2.48 × 10^13 PFU/mL) or p53 (3.23 × 10^13 PFU/mL) (Vector Builder) under cell-specific cre promoters. AgRPp53 KO and AgRP-Cre mice fed a HFD for 10 weeks before to viral injections were anesthetized and placed in a stereotaxic frame (Kopf Instruments). Body weight and food intake were recorded until 4 weeks after the surgery. Specific overexpression of p53 in AgRP neurons was evaluated by immunohistochemistry.

**i.c.v. treatments.** i.c.v. surgery was carried out on 16-week-old male AgRPp53 KO mice, in littermate controls, and in 8-week-old rats. Mice were anaesthetized with a ketamine/xylazine cocktail (ketamine 15 mg/kg BW/xylazine 3 mg/kg BW) and i.c. v. cannulae were implanted stereotaxically (David Kopf Instruments, Tujunga, CA), in the lateral ventricle[73]. The coordinates used were −0.6 mm posterior to bregma, ±1.2 mm lateral from midline, and −2.0 mm ventral from the brain surface for mice. For rats we used the following coordinates: −0.9 mm posterior to bregma, ±1.5 mm lateral from midline, and −3.5 mm ventral from the brain surface, as described previously[38,70,72,74]. After surgery, the animals were singly housed and given at least 4 days to recover. AgRPp53 KO mice were infused with 2 μl of either vehicle (saline), or TUDCA (2.5 μg/μl) (Calbiochem) for 7 consecutive days (once a day, 1 h before lights off), as described previously[38]. In a separate experiment AgRPp53 KO^P mice were infused with 2 μl of either vehicle (DMSO) or JNK inhibitor (SP-600125; 1.5 μg/mice)[44] for 6 consecutive days (once a day, 1 h after the light on). Food intake and body weights were recorded daily. After the pharmacological treatment or vehicle injection, mice were either culled to dissect the hypothalamus studies. Finally AgRPp53 KO mice were infused with 2 μl of either vehicle (saline), or ghrelin (2.5 μg/μl) (Bachem) for 2 consecutive days (once a day, 1 h after lights on) food intake and body weights were recorded daily and at 2, 4, and 6 h after the first injection.

Rats were anesthetized by an i.p. injection of ketamine/xylacine cocktail mixture (ketamine 100 mg/kg BW/xylazine 15 mg/kg BW). Brain infusion cannulae were stereotaxically placed into the lateral ventricle or directly into the ARC or VMH (Plastic One, Roanoke, VA, USA). Rats were anesthetized and cannulate to an osmotic minipump (model 2001 Alzet Osmotic Pumps; DURECT, CA) as previously described[73]. The minipump was inserted in subcutaneous pocket on the dorsal surface. The incision was closed with sutures, and rats were kept warm until full recovery. Rats were infused with either vehicle (saline) or p53 agonist (adriamycin 1 μg/rat/day) along the time of the experiments. To test the contribution of the SNS, a specific antagonist of the β3-adrenergic receptor, SR59203A (3 mg/kg/day; Tocris Bioscience Bristol), was administrated subcutaneously at the beginning and at the end of the light cycle during the treatment with adriamycin[24,71].

**Electrophysiology.** Perforated patch clamp recordings were performed on AgRP neurons in brain slices from male AgRP-Cre and AgRPp53 KO mice fed a HFD for 10 weeks and injected in the mediobasal hypothalamus with AAV-DIO-hSyn-EGFP that expressed GFP specifically in AgRP neurons. Experiments were carried out essentially as described previously[75]. Briefly, AgRP neurons were identified according to their anatomical location in the ARC and reporter expression. Brain slices were continuously superfused with carbogenated artificial cerebrospinal fluid (aCSF) (~31 °C) at a flow rate of ~2 ml/min (recording chamber volume: ~2 ml). aCSF contained (in mM): 125 NaCl, 2.5 KCl, 2 MgCl₂, 2 CaCl₂, 1.2 NaH₂PO₄, 21 NaHCO₃, 10 HEPES, 5 glucose, adjusted to pH 7.2 with NaOH, resulting in an osmolarity of ~310 mOsm. Recordings were performed with pipette solution containing 1% biocytin (Sigma) and (in mM): 128 K-gluconate, 10 KCl, 10 HEPES, 2 MgCl₂, and adjusted to pH 7.3 with KOH. Initially, the patch pipette was tip filled with internal solution and back filled with 0.02% tetraethylrhodamine-dextran (D3308, Invitrogen, Eugene, OR, USA) and Amphotericin B-containing internal solution (~200 μg ml$^{-1}$; A4888, Sigma) to achieve perforated patch recordings[76,77]. Amphotericin B was dissolved in DMSO (D8418, Sigma) as described previously[78] and was added to the modified pipette solution shortly before use. To block GABAergic and glutamatergic synaptic input the aCSF contained 10$^{-4}$ M picrotoxin (P1675; Sigma-Aldrich), 5 × 10$^{-5}$ M D-AP5 (A5282; Sigma-Aldrich), and 10$^{-5}$ M CNQX (C127; Sigma-Aldrich). During the perforation process access resistance ($R_a$) was constantly monitored and experiments were started after $R_a$ had reached steady state (~15–20 min) and the action potential amplitude was stable. To confirm the integrity of the perforated patch recording, $R_a$ was monitored over the course of the experiment. A change to the whole-cell configuration was also indicated by Dextran-Fluorescein fluorescence in the cell body.

**Neuron labeling.** After the electrophysiological experiments, perforated-patch recordings were converted to the whole-cell configuration and biocytin was allowed to diffuse into the cell for at least 1 min. Biocytin-streptavidin labeling combined with GFP immunohistochemistry was performed as previously described[75].

**Data analysis.** Data analysis was performed with Spike2 (CED), Igor Pro 6 (Wavemetrics), and Graphpad Prism (version 5.0c; Graphpad Software Inc.). Data are given as mean ± standard error. To determine differences in means between genotypes, unpaired t-tests were performed. A significance level of 0.05 was accepted for all tests. N-values and p-values are reported as exact numbers above the bar graphs.

Excitability: To analyze excitability, i.e., evoked action potential firing, a series of depolarizing current pulses (5–25 pA in 5 pA increments; 5 s duration) were injected. For each current pulse the number of action potentials was determined.

SFA: To determine SFA ratios 5 s depolarizing stimuli were applied from a holding potential of $\sim{-}70$ mV with initial instantaneous AP frequencies between 30 and 35 Hz. Instantaneous frequencies were plotted over the 5 s time course and fit to a mono-exponential decay equation with $Y_0$ set to the initial instantaneous frequency: $Y = (Y_0 - \text{Plateau}) \cdot \exp(-K \cdot X) + \text{Plateau}$, where Plateau is the asymptotic frequency, $K$ is the inverse time constant, and $T$ is the time.

**Western blot analysis**. Western blot was performed as previously described[72,74]. Briefly, animals fed ad libitum were sacrificed and protein lysates from ARC (20 μg) and MBH (20 μg) were subjected to sodium dodecyl sulfate–polyacrylamide gel electrophoresis, electrotransferred on a polyvinylidene difluoride membrane, and probed with different antibodies (Supplementary Table 1). For protein detection we used horseradish peroxidase-conjugated secondary antibodies (Dako Denmark, Glostrup, Denmark) and chemiluminescence (Pierce ECL Western Blotting Substrate; Thermo Scientific, Waltham, MA). Then, the membranes were exposed to radiograph film (Super RX, Fuji Medical X-Ray Film; Fujifilm, Tokyo, Japan) and developed with developer and fixing liquids (AGFA, Mortsel, Belgium) under appropriate dark room conditions. The protein levels were normalized to β-actin or α-tubulin for each sample. Uncropped blots are shown in Supplementary Fig. 16.

**Immunofluorescence**. Rats and mice brains were fixed by perfusion followed by immersion (12 h) in 10% buffered formalin for 24 h. The brain pieces were cut 50 μm thick using a Vibratome® Series 1000. Detection of GFP, fluorescein isocthiocyanate, TOMATO immunofluorescence, and double labeling were performed as previously reported[71] using a mouse anti-p53 antibody (Cell Signaling Technology 2524).

**Real-time PCR**. RNA was extracted using Trizol® reagent (Invitrogen) according to the manufacturer's instructions. 2 μg of total RNA were used for each RT reaction, and cDNA synthesis was performed using the SuperScript™ First-Strand Synthesis System (Invitrogen) and random primers (Supplementary Table 2) as previously described[24].

**Hematoxylin/eosin staining and immunohistochemistry**. WAT, BAT, and liver samples were fixed in 10% formalin buffer for 24 h, and then dehydrated and embedded in paraffin by a standard procedure. Sections of 3 μm were prepared with a microtome and stained using a standard Hematoxylin/Eosin Alcoholic (BioOptica) procedure according to the manufacturer's instructions[73]. Alternative sections of paraffin were used for immunohistochemistry detection of UCP-1. Immunohistochemistry was performed as described[24,79] using a rabbit anti-UCP-1 (1:2000; Abcam; Cambridge, UK). UCP-1-positive cells were counted by using *Frida Software* (the Johns Hopkins University; Baltimore, MD, USA). Up to 10 animals per experimental group were used and 3 pictures per each image sections were analyzed.

**AgRP immunohistochemistry and fiber density analysis**. Mice were transcardially perfused (4% paraformaldehyde). Brains were cryoprotected, frozen in smashed dry ice, and subsequently sectioned using a freezing microtome. Selected 30 mm sections (one of every four sections) were blocked with 2% chicken serum in KPBS + 0.4% Triton X-100 and incubated with rabbit anti-AGRP antibody (1:500; Phoenix Pharmaceuticals), rabbit anti-GFAP (Dako; Agilent), or rabbit anti-iba1 (ionized calcium-binding adapter molecule 1; Synaptic Systems)[80] in blocking solution for 72 h at 4 °C. As secondary antibody, a chicken anti-rabbit Alexa Fluor 488 (1:300; Life Technologies) in KPBS + 0.4% Triton X-100 was used (2 h at room temperature). For quantification, representative sections through the PVH (bregma between −0.59 and −1.23 mm) and ARC (bregma between −1.43 and −1.91 mm) of each animal were acquired using a Leica DMI 4000B confocal microscope equipped with a 20× objective (numerical aperture 0.70). Ten image stacks with 1 μm distance interval throughout the PVH and ARC of each animal were taken. AgRP fiber density analysis was then performed using ImageJ Launcher and based on previously published reports[81]. Briefly, each single image was binarized to compensate for differences in fluorescence intensity, specified in a random 200 × 200 μm region and skeletonized, so that each fiber segment was 1 pixel thick. The integrated intensity was then measured for each image. The total density value was obtained by the sum of all image planes analyzed.

**Data analysis and statistics**. Data are expressed as mean ± SEM. mRNA and protein data were expressed in relation (%) to control (vehicle-treated) rats. Error bars represent SEM. Statistical significance was determined by Student's $t$-test when two groups were compared or ANOVA and post hoc two-tailed Bonferroni test when more than two groups were compared. $P < 0.05$ was considered significant.

**Data availability**. All data generated or analyzed during this study are included in this published article (and its Supplementary Information files) or from the authors upon reasonable request.

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

## Acknowledgements

This work has been supported by grants from Ministerio de Economia y Competitividad (C.D.: BFU2011-29102; M.L.: SAF2015-71026-R; and BFU2015-70454-REDT/*Adipoplast*; G.S.: SAF2016-79126-R; R.N.: BFU2015-70664R), Xunta de Galicia (M.L.: 2015-CP079 and 2016-PG068; R.N.: 2015-CP080 and 2016-PG057), Atresmedia Corporación (M.L.), Astra Zeneca Fundation (R.N.), the European Foundation for the Study of Diabetes (G.S. and R.N.), DFG grant SFB 1218/TP B07 (P.K.); Helse Vest RHF (J.F.; the Western Norway Regional Health Authority); Comunidad de Madrid IMMUNOTHERCAN-CM (B2017/BMD3733) (G.S.), Fundación BBVA (G.S.). Centro de Investigación Biomédica en Red (CIBER) de Fisiopatología de la Obesidad y Nutrición (CIBERobn). CIBERobn is an initiative of the Instituto de Salud Carlos III (ISCIII) of Spain which is supported by FEDER funds. The research leading to these results has also received funding from the European Community's Seventh Framework Programme under the following grant: ERC StG-2011-OBESITY53-281408 to R.N. M.Q. is a recipient of a Postdoctoral fellowship from Galician Government (Xunta de Galicia ED481B2014/039-0). R.H.-T. is a recipient of a postdoctoral fellowship FAPESP (Process: 2016/01868-2). O.A.-M. is funded by the ISCIII/SERGAS thought a research contract "Sara Borrell" (CD14/00091). E.S.-R. is recipient of a Predoctoral fellowship from Ministerio de Economia y Competitividad (ref: BES-2013-062796).

## Author contributions

M.Q., O.A.-M. C.F. carried out the experiments, analyzed data, and developed analytical tools and contributed to discussion. S.B., R.G., L.T.-L., R.H.-T., C.G.-C., R.H.B., B.Y.H.L., D.B., E.S.-R., A.S., J.A.M., P.V., M.F.-F., R.C., P.P. carried out the experiments and analyzed data. J.F., M.M.M., J.C.B., G.Y., M.-T., C.D., M.L., M.C., P.K. and G.S.

contributed to discussion and designed experiments. M.Q., O.A.-M. and R.N. designed experiments and wrote the paper. R.N. serves as the guarantor.

## Additional information

**Competing interests:** The authors declare no competing interests.

