## [Peer Review File · Nature Communications]

Reviewers' comments:

Reviewer #1 (Remarks to the Author):

Quiñones et al. in their manuscript entitled "p53 in AgRP neurons is required for protection against diet-induced obesity via JNK1" shows a significant role of p53 in the regulation of food intake, bodyweight and metabolic parameters through its action in the AgRP neurons.

Authors have performed elegant studies to investigate the role of p53 in the hypothalamus. First they have inhibited the action of p53 in the ARC and VMH. The results obtained from these studies indicated that p53 could be playing a role in the ARC for regulation of food intake and bodyweight. Subsequently, authors have used POMC and AgRP-Cre models to address which neuron populations is mediating the role of p53 and discovered that p53 in AgRP neurons are responsible for the observed effects. In addition, authors have used various chemical approaches including reducing ER stress with TUDCA, inhibition of JNK and finally activation of p53 with Adriamycin in the hypothalamus to further dissect the mechanism.

Detailed results provided in this MS supports the hypothesis of authors and bring a novel insight to obesity field.

I have only minor comments on the paper;

- 1) The last part of the paper, which is related to Ghrelin action is not that much related the rest of the paper including the ER stress and JNK connection. Authors should, in my opinion, remove this part from this MS or at least take it into supplementary part.
- 2) On page 9 of the MS authors indicate that JNK activation precedes changes in hypothalamic ER stress. Their results also show that reducing ER stress by TUDCA also leads to a reduction in JNK activity. This point should be discussed in the discussion part in detail.

Reviewer #2 (Remarks to the Author):

Major remarks:

1-The authors performed several experiments to prove that p53 in AgRP neurons influences weight gain and that targeting p53 results in significant body weight changes. While the experiments performed are informative, the paper remains very descriptive. The authors propose that in AgRP neurons p53 functions via JNK1, as decreased p53 results in increased p-JNK. However, they do not explain how p53 regulates p-JNK levels.

2-In the rat system, inhibition of p53 is achieved by expression of a DNp53RK – which seems to be a dominant negative kinase (maybe RET) that is thought to inhibit the activation of p53. As I understand, this is not a dominant negative form of p53 – an approach that has been more commonly used. I do not believe this is a reliable way to inactivate p53, and indeed the authors provide no evidence that p53 activity is diminished using this approach. I think the data derived from this model would need strong verification using a system that effectively and specifically depleted p53.

3- The authors then move to a mouse system in which p53 can be deleted using Cre in POMC or AgRP

neurones. Again, the data supporting the deletion of p53 is not highly compelling. What happens to p53 levels in the floxed mice after HFD (eg comparison to wild type shown in Figure S2B)?

4-The authors reported increased p53 levels in the ARC in both mice and rats in response to prolonged (12 weeks) HFD feeding. However they also claim that p53 levels are unchanged if assessed after a shorter (2-4 weeks) exposure to HFD. However, changes in weight in the AgRP-Cre p53 fl/fl mice are seen from 8-10 weeks. Obesity is the result of chronically positive energy balance. So mechanisms that promote obesity happen before obesity is manifested. The authors should investigate further how p53 is modulated in the early phases of HFD exposure. Are phosphorylation or acetylation levels different?

5- There is no evidence in the study that p53 is activated under these conditions. The authors would need to show induction of p53 target genes.

4- In fig 5 the authors show that P-JNK levels are up-regulated in AgRP-cre p53flox/flox mice after short exposure (3 weeks) of HFD. This experiment is very informative, but unfortunately some important parameters are not reported. In particular, it would be important to report the total food intake, weight gain in those 3 weeks in addition, what about p53 level and status? And finally, is energy expenditure already affected? If p53 regulates JNK, thermogenesis and by doing so, weight gain, differences in thermogenesis must be detected at very early time points.

5- AgRP-cre p53flox/flox mice display increased food intake only in response to HFD but not to chow diet (fig 3b). This result is of particular interest because it suggests that p53's role in controlling food intake is specific to HFD feeding. Unfortunately, this observation was not investigated any further, but understanding if (and how) p53 responds to excessive caloric intake per se or possible specific nutrient would greatly increase the novelty of this work.

7- given the role of inflammation in the control of food intake and energy expenditure, (Garcia-Caceres C, 2013, Solinas 2012) it is surprising that the authors did not measure markers of inflammation in the brain. This is particularly important in studies regarding p53, given its role in stress response.

8- In Figure 6, the authors turn to ectopic overexpression of p53 using adenovirus expressed p53. Unfortunately, massive overexpression of p53 is likely to drive responses that would be detrimental to the function of cells (such as death and senescence) that may not be reflective of the normal function of endogenous p53, so these overexpression experiments can be very misleading. Again, there is no investigation of p53 activity and how this compared to the activity of p53 when diet is modulated. The use of adriamycin is similarly complicated by the numerous other effects of this drug. Possibly the use of a more selective activator of p53 (like Nutlin) would be more informative.

Minor comments

1- in GTT and ITT absolute values and not percent should be reported.

2- Blood glucose levels in the ITT performed drops quite dramatically. This suggests that the concentrations used are too high and may have masked some phenotype.

3- The authors claim that only 80% of AgRP or POMC neurones express p53 – it would be more accurate to say only 80% of these cells express levels of p53 that can be detected by immunohistochemistry. I do not think they can conclude (as they do in the discussion) that some of

these cells do not express p53.

5- Fig 3 g-h: given the ongoing discussion about how to properly normalise EE data, I would invite the author to report results not normalized on grams as supplementary figure. (Tschoep M. H 2012)

6- Given the role of p53 in cell death and survival, the authors should investigate if AgRP neurons have a different sensitivity to the JNK inhibitor they used. Brain sections after the experiment with the inhibitor displaying similar neuronal population, inflammation and cell death are probably required.

7- Fig 8: it is unclear if animals were exposed to HFD or to chow diet. However, AgRP-cre p53flox/flox mice display similar food intake on chow diet and even increased food intake on HFD. This would suggest, in a very simplistic manner, a normal or increased sensitivity to Ghrelin. It is possible however that AgRP-cre p53flox/flox are overcompensating ghrelin resistance by more circulating ghrelin levels, which should be measured by ELISA.

8-It is surprising that the authors did not measure the activity of hypothalamic-pituitary- thyroid axis as it was reported to be involved in the metabolic regulation by JNK (Sabio 2010).

9-It is unclear if the circadian pattern of food intake is affected in AgRP-cre p53flox/flox, especially in response to HFD. Alteration of circadian pattern of feeding has been associated with obesity (Tureck 2005) and p53 has been reported to interact with the circadian clock (Miki 2013).

10-It is not clear for how many hours animal have been food-deprived before sampling for protein analysis.

Reviewer #3 (Remarks to the Author):

In this manuscript, Quinones et al. examine the role of p53 in the medial basal hypothalamus in energy balance control. Using a combination of mouse and rat models, these researchers show that reduction of p53 in the medial basal hypothalamus and in AgRP neurons specifically increases sensitivity to high fat diet induced obesity secondary to increased feeding and decreased energy expenditure. This is associated with increase JNK activity and central pharmacologic inhibition of JNK is sufficient to reverse this increased sensitivity to obesity. Moreover, overexpression of p53 in the hypothalamus of mice on a high fat diet decreases JNK activity, food intake and energy expenditure. Based on these data, the authors suggest that p53 activity in AgRP neurons is required for the adaptation to diet-induced obesity.

Overall this is an interesting hypothesis and novel data implicating p53 in the regulation of energy balance. These observations expand our understanding of the molecular mechanisms contributing to energy balance control by the central nervous system. The investigation of both p53 knock down as well as p53 overexpression strengthens paper as a whole. However, I do feel that the following areas need to be addressed in more detail.

Concerns/questions:

1. p53 regulates gene transcription and cell cycle control. Whereas the authors link the loss of p53 to changes in ER stress and JNK activity there is little data or explanation for how p53 activity may mediate these changes observed in the central nervous system. One could argue that loss of p53 will have pleiotropic effects on cell function leading to a range of cellular and physiologic derangements. Given the proposal that the altered physiology seen with p53 deletion is mediated predominantly by the AgRP neuron, studies examining the electrophysiologic function of the AgRP neuron are needed

(esp for publication in a high profile journal). In addition, as loss of p53 might lead to developmental changes in AgRP neurons, confirming appropriate AgRP projection targeting/densities is warranted.

2. The immunohistochemical analysis of p53 and its manipulation is underwhelming (supplemental Figure 3). P53 is poorly visualized and AgRP-Cre activity (as read out by a tomato reporter) is much broader than expected, suggesting ectopic Cre activity in this model. Given that the authors focus their attention on a critical role of p53 in AgRP neurons, additional approaches confirming the loss of p53 only to AgRP neurons are indicated. And again, given the potentially pleiotropic effects of p53 on cell activity, cell counts of AgRP-loxp53 vs AgRP-reporters should be performed to confirm that the number of AgRP cells is similar between groups.

3. Figure 1 documents the effects of p53 inhibition by injection of a dominant negative p53 adenoviral vector. Although there is a representative image provided, the authors need to estimate the number of cells transduced by this approach and give some sense of the injection location. Again this is important because the authors have implicated the AgRP neuron as the critical neuronal player in the phenotypes they have uncovered. Are animals with fewer ARC cells transduced less affected by the loss of p53? Are the ARC cells transduced actually expressing AgRP?

4. I am slightly concerned that the bodyweight differences detected following ARC injection of the dominant-negative p53 are really a reflection of an unexpected flattening of weight gain in the control group as opposed to increased weight gain in the p53 inhibited group (Figure 1C). Similarly, the phenotypes of the AgRP-cre, lox-p53 mice in Figure 3 is somewhat confusing. Whereas the lines suggest a difference in bodyweight on HFD for control and AgRP-cre lox p53 groups, there is a significant flattening of bodyweight gain in the controls at week 16-18 that enhances the effect; interestingly, it is only at this point when a clear food intake difference emerges. This needs some explanation, esp if the phenotypic analysis of these mice is performed after the flattening of the controls is observed.

5. The authors demonstrate that increased activation of p53 (pharmacologically or genetically) in the ARC can ameliorate diet-induced obesity. These approaches support their hypothesis that p53 activity can drive changes in overall energy balance. These approaches are not limited to manipulating p53 action in the AgRP neuron only; if the authors' central thesis is that p53 action in AgRP neurons is of critical importance, overexpression of active p53 in AgRP neurons specifically would be a more direct test of this question.

Reviewer #4 (Remarks to the Author):

The manuscript by Quinone and colleagues examines the role of p53 in AgRP neurons in diet-induced obesity (DIO). They report that mice lacking p53 selectively in AgRP, but not POMC or VMN neurons are more susceptible to DIO. This effect is due, in part, to increased energy intake and reduced energy expenditure, and associated with increased hypothalamic JNK activity. In addition, they suggest that pharmacological inhibition or genetic deletion of JNK reverses the obesity phenotype of mice lacking p53 in AgRP neurons. Taken together, the authors conclude that p53 in AgRP neurons is required for normal adaptation against DIO. Overall, the findings are of interest, although several points require additional clarification.

The increased susceptibility to DIO in mice lacking p53 in AgRP neurons (KO mice), but not POMC or VMN neurons on a HFD is compelling. However, evidence that inhibition of p53 in the ARC increases food intake and body weight in chow-fed rats is relatively small and less convincing. More importantly, how do the authors interpret these findings given that mice lacking p53 in the ARC on a chow-diet do not exhibit a body weight phenotype. Was this study also performed in rats on a HFD? Finally, there is also concern that this viral approach is not selective to the ARC, with expression detected along the walls of the 3V (Figure 1A).

Mice lacking p53 in AgRP neurons exhibit increased body weight when exposed to a HFD. It is recommended that food intake (Fig 3B) be converted to kcal/week instead of grams. In addition, non-fat mass should be presented as grams, rather than as a %BW. This can be misleading, particularly as the KO mice appear to have elevated levels of non-fat mass (Fig 3H). Are the animals bigger (organ weight, head to anus length) as well as more obese? Given the difference in size of the animals, it is also recommended the multiple linear regression is performed on energy expenditure data (see MMPC website).

The authors also report that the KO mice are more insulin sensitive. This study should be either removed or repeated. Given the blood glucose levels drop below 30mg/dl in both WT and KO mice, this study examines more the response to hypoglycemia, rather than insulin sensitivity. Thus, to test this question, a lower dose of insulin is required.

In the subsequent studies, the authors examine the effects on ER stress and JNK in WT and KO mice on a HFD. A subtle, but important point that requires attention and is important in the interpretation, is that previous evidence suggests that ER stress and JNK is increased in HFD-fed relative to chow-fed WT mice. Thus, statements throughout the ms are suggested to be revised accordingly. For example (line 222) to read these results suggest that "further" activation.... In addition, studies using the ER stress reliever (TUDCA; Fig. 4) or the JNK inhibitor SP-600125 (Fig. 5) may also benefit from a chow-fed control group. Overall, the findings on body weight from both of these studies are a little underwhelming, particularly as the control-saline-treated group lose considerable body weight. Moreover, given that HFD induces ER stress and JNK, based on the previous literature, one would predict that these treatments would induce weight loss in HFD WT mice, but the weight loss would have been exacerbated in the KO mice. This warrants further discussion. Food intake is also best presented as mean daily energy intake, rather than cumulative energy intake.

The color scheme for the Legend in Figure 5L needs to be revised.

To determine whether pharmacological activation of p53 in the ARC is sufficient to attenuate food intake and body weight gain, animals receive Adriamycin (a drug activating p53) directly into the ARC using osmotic minipumps. There are major concerns with this approach, given that although the cannula is directed to the ARC, the minipump delivers 1ul/hr (24ul/d) and will not only leak into adjacent hypothalamic nuclei, but into the 3V. Thus, the p53 can act anywhere throughout the brain. The authors provide data that delivery into the VMH infusion of Adriamycin does not have any feeding or body weight effects, although this cannot be fully interpreted as the authors only examine food intake and body weight over 5 days, a time when there was no effect following ARC administration.

The authors provide evidence that in mice lacking p53 in AgRP neurons exhibit a blunted response to ghrelin. Are these in chow or HFD-fed mice? Can the authors discuss how this observation might contribute to the increased susceptibility to DIO?

Reviewer #1 (Remarks to the Author):

Authors have performed elegant studies to investigate the role of p53 in the hypothalamus. First they have inhibited the action of p53 in the ARC and VMH. The results obtained from these studies indicated that p53 could be playing a role in the ARC for regulation of food intake and bodyweight. Subsequently, authors have used POMC and AgRP-Cre models to address which neuron populations is mediating the role of p53 and discovered that p53 in AgRP neurons are responsible for the observed effects. In addition, authors have used various chemical approaches including reducing ER stress with TUDCA, inhibition of JNK and finally activation of p53 with Adriamycin in the hypothalamus to further dissect the mechanism.

Detailed results provided in this MS supports the hypothesis of authors and bring a novel insight to obesity field.

RESPONSE: We would like to thank the Reviewer for the positive and encouraging comments on our manuscript.

I have only minor comments on the paper;

1) The last part of the paper, which is related to Ghrelin action is not that much related the rest of the paper including the ER stress and JNK connection. Authors should, in my opinion, remove this part from this MS or at least take it into supplementary part.

RESPONSE: We understand the point of the Reviewer and we admit that this figure is not strictly related with ER stress or JNK. However, these results aim to demonstrate that in addition to its role in conditions of nutrient excess, p53 in AgRP neurons is also of physiological relevance for the actions of endogenous hormones, such as ghrelin. We believe that, considering the essential role of AgRP neurons as mediators of the metabolic actions of ghrelin, those data are important. Supporting this view, another Reviewer has suggested to measure circulating levels of acyl-ghrelin and check the possibility that mice lacking p53 in AgRP neurons might be ghrelin-resistant; therefore, we have decided to maintain that Figure (*Nature Communications* allows 10 figures). Nevertheless, we leave this decision at the discretion of the Editor and the Reviewer; if both of you still find this information unnecessary, we will be happy to remove it.

2) On page 9 of the MS authors indicate that JNK activation precedes changes in hypothalamic ER stress. Their results also show that reducing ER stress by TUDCA also leads to a reduction in JNK activity. This point should be discussed in the discussion part in detail.

RESPONSE: We apologize if we did not explain this part correctly, but there is some kind of misunderstanding here. The first assumption is indeed true, JNK activation precedes changes in hypothalamic ER stress, however we did not measure JNK or pJNK in the mediobasal hypothalamus of mice treated with TUDCA (former Figures 4C-4H, **new Figures 5C-5I**).

Reviewer #2 (Remarks to the Author):

Major remarks:

1-The authors performed several experiments to prove that p53 in AgRP neurons influences weight gain and that targeting p53 results in significant body weight changes. While the experiments performed are informative, the paper remains very descriptive. The authors propose that in AgRP neurons p53 functions via JNK1, as decreased p53 results in increased p-JNK. However, they do not explain how p53 regulates p-JNK levels.

RESPONSE: We thank the reviewer for the comments and interesting questions that have been addressed as described below. In this study we have generated 3 conditional knockouts (mice lacking p53 in SF1, POMC and AgRP neurons), performed virogenetic approaches and described that JNK1 (using JNK inhibitor and JNK1 ko mice as tools) is essential for the metabolic actions of p53 in AgRP neurons. This is the first study describing that brain p53 exerts a regulatory role in metabolic homeostasis. As expected, and in keeping with most manuscripts reporting something for first time, some new questions come out, and we have an almost unlimited number of mechanistic studies worth undertaking. In this regard we include in the new version of this manuscript some new data uncovering the role of MKK4 and MKK7 as downstream signals of p53 in our experimental paradigm. In sum, with all due respect, we strongly believe that our paper novel is a novel insight on how p53 may influence energy homeostasis and identify hypothalamic AgRP neurons as the main loci of action. We further documented that this effect is largely due to alterations in energy expenditure. The novelty and wealth of information provided is without any doubt up to the high standard usually required by Nat Comms and concluding that the paper is just informative and descriptive is from our point of view an underestimation of this work.

We agree with the reviewer that it is important to address how p53 regulates p-JNK levels. To address this matter, we have measured MKK4 and MKK7 (the upstream kinases of JNK) (Davis RJ. *Cell*. 2000). Our results indicate that both MKK4 and MKK7 protein levels are elevated in the mediobasal hypothalamus of AgRP-Cre; p53^{loxP/loxP} mice fed a HFD during 13 weeks and sacrificed at week 18 in comparison to their controls (**new Figure 5B**). However, when mice were fed a HFD for only 3 weeks, we found that only MKK7 was upregulated in the mediobasal hypothalamus of AgRP-Cre; p53^{loxP/loxP} mice (**new Figure 6J**). This implies that increased phosphorylation of pJNK in AgRP-Cre; p53^{loxP/loxP} mice after 3 weeks on HFD is specifically associated to MKK7. Next, we also measured both MKK4 and MKK7 in the mediobasal hypothalamus of DIO mice where p53 was over-expressed in the ARC; our findings indicate that protein levels of both MKK4 and MKK7 are decreased in the mediobasal hypothalamus of mice with p53 over-expression compared to their wildtypes (**new Figure 8O**). Altogether, these results suggest that p53 regulates the expression of both MKK4 and MKK7 and that subsequently phosphorylates JNK.

2-In the rat system, inhibition of p53 is achieved by expression of a DNp53RK – which seems to be a dominant negative kinase (maybe RET) that is thought to inhibit the activation of p53. As I understand, this is not a dominant negative form of p53 – an approach that has been more

commonly used. I do not believe this is a reliable way to inactivate p53, and indeed the authors provide no evidence that p53 activity is diminished using this approach. I think the data derived from this model would need strong verification using a system that effectively and specifically depleted p53.

RESPONSE: We totally agree with the Reviewer that verification of this viral vector is essential. As stated in the previous version, we have tested this viral vector in a previous publication (reference 24: Al-Massadi O et al, Endocrinology 2016, Figure 4, please see below). Double-mutant kinase dead p53 regulating kinase (DNTp53RK) acts as a dominant negative (the TP53RK clone was then mutated at g217a and a218t to E73M), which blocks p53 activation and therefore function. Since we confirmed that this system is indeed inhibiting p53, we did not include this information in the present manuscript to avoid redundancy.

[Redacted]

[Redacted]

3- The authors then move to a mouse system in which p53 can be deleted using Cre in POMC or AgRP neurones. Again, the data supporting the deletion of p53 is not highly compelling. What happens to p53 levels in the floxed mice after HFD (eg comparison to wild type shown in Figure S2B)?

RESPONSE: We have performed additional studies to address this question. As the reviewer may be aware, p53 is widely expressed in most of the cells. Thus, we investigated which percentage of AgRP neurons express p53. FACS sorting and single-cell RNA sequencing of AgRP-eGFP neurons showed that 31 out of 45 AgRP neurons express p53 (GEO Database repository: GEO Accession: GSE92707) (**new Supplementary figure 4A**). In addition, we now provide higher magnification photos showing that AgRP-cre p53^{fl/fl} and POMC-cre p53^{fl/fl} have indeed a much lower colocalization of p53 and Tomato/AgRP than their control littermates (**new Suppl. Figs 4B and 4C**).

To see what happens to p53 levels in mice fed a chow diet or a HFD for 13 weeks (former Suppl. Figure 2B, **new Suppl. Figure 3B**) we have measured downstream target genes of p53 such as Bax and p66shc (Chipuk JE et al, *Science* 2004, Tomita K et al. *J Hepatology* 2012) (**new Suppl. figure 3C**). We found that according to the increased p53 expression, protein levels of Bax and p66-shc are up-regulated in the mediobasal hypothalamus of mice fed a HFD compared to mice fed a chow diet. These data imply that HFD triggers p53 activation in the mediobasal hypothalamus.

4-The authors reported increased p53 levels in the ARC in both mice and rats in response to prolonged (12 weeks) HFD feeding. However they also claim that p53 levels are unchanged if assessed after a shorter (2-4 weeks) exposure to HFD. However, changes in weight in the AgRP-Cre p53 fl/fl mice are seen from 8-10 weeks. Obesity is the result of chronically positive energy balance. So mechanisms that promote obesity happen before obesity is manifested. The authors should investigate further how p53 is modulated in the early phases of HFD exposure. Are phosphorylation or acetylation levels different?

RESPONSE: This is a very good point. However, we believe that regulation of p53 and the biological effect of p53 are two different things that do not necessarily correlate in time. Mice lacking p53 in AgRP neurons gain more weight than their controls at week 10 (Figure 3A), but this does imply that the expression of p53 must be altered in the hypothalamus of DIO mice at this specific time point. p53 expression and activity are markedly influenced by cellular stress, damage, hypoxia, etc.; thus, it seems plausible that HFD needs time to alter cellular metabolism. Thus, from our point of view, it is not so surprising that p53 is altered after 13 weeks of HFD but not at earlier phases of HFD exposure. However, and as explained in point 6, mice lacking p53 in AgRP neurons fed a HFD for 3 weeks showed increased levels of MKK7 and phosphorylated JNK in the mediobasal hypothalamus (**new Figure 6**), supporting that these molecular events precede obesity.

Following the recommendation of the reviewer, we have measured phosphorylated and acetylated levels of p53 in the mediobasal hypothalamus of mice fed a chow diet or HFD during 2 and 4 weeks, but we failed to find differences between these groups (please see Appendix 2 below). In addition, we have also measured Bax, a p53 target gene that is involved in some of the p53-related metabolic actions in other tissues (Chipuk JE et al, *Science* 2004). As shown below, and consistent with phosphorylated and acetylated levels of p53, BAX protein levels remained unchanged after 2 and 4 weeks of HFD. In summary, p53 expression requires long exposure to HFD, but the lack of p53 in AgRP neurons induces biochemical changes, meaning an up-regulation of MKK7 and pJNK in the mediobasal hypothalamus before changes in adiposity, BAT temperature and body weight can be detected (**new Figure 6**, please see also response to point 6).

Appendix figure 2. Protein levels of acetyl-p53, phosphor-p53 and BAX in the mediobasal hypothalamus of mice fed a HFD for 2 and 4 weeks.

5- There is no evidence in the study that p53 is activated under these conditions. The authors would need to show induction of p53 target genes.

RESPONSE: Again, we agree with the reviewer that this is a good point and in addition to p53 levels, we have also measured protein levels of several p53 target genes like BAX and p66-shc, which have been shown to be involved in some of the p53-related metabolic actions in other tissues (Chipuk JE et al, *Science* 2004)(Tomita K et al. *J Hepatology* 2012). Our findings indicate that protein levels of BAX and p66-shc are increased in the mediobasal hypothalamus of mice fed a HFD during 13 weeks in comparison to mice fed a chow diet (**new Supplementary figure**

3C). Thus, these new results support the hypothesis that hypothalamic p53 is activated under conditions of HFD.

6- In fig 5 the authors show that P-JNK levels are up-regulated in AgRP-cre p53flox/flox mice after short exposure (3 weeks) of HFD. This experiment is very informative, but unfortunately some important parameters are not reported. In particular, it would be important to report the total food intake, weight gain in those 3 weeks in addition, what about p53 level and status? And finally, is energy expenditure already affected? If p53 regulates JNK, thermogenesis and by doing so, weight gain, differences in thermogenesis must be detected at very early time points.

RESPONSE: We agree that this information is relevant. Weekly and total food intake (in grams and kcal), body weight, body weight change, and results of the indirect calorimetry system, as well as thermography and UCP1 protein levels in BAT have been added to the manuscript (**new Figures 6A-6I**). These findings show that all these parameters and measurements were very similar between control and mice lacking p53 in AgRP neurons fed a HFD for 3 weeks. Despite the similar body weight, adiposity, food intake and energy expenditure at this time point, the lack of p53 is increasing MKK7 and phosphorylated JNK in the mediobasal hypothalamus of mice fed a HFD for 3 weeks, supporting that these molecular events precede changes in thermogenesis and energy expenditure, thereby also preceding changes in body weight. We have a large experience performing measurements with thermography and the indirect calorimetry system (Whittle et al, Cell 2012)(Martinez de Morentin, Cell Metab, 2014)(Matesanz N et al, Nat Commun 2017)(Martínez-Sánchez N et al, Cell Metab 2017)(Ramirez S et al, Cell Metab 2017), etc. and in our hands, differences in thermogenesis and specially energy expenditure (or respiratory quotient in other cases) are only robust when changes in body weight are evident. However, changes at biochemical level can indeed be detected before differences in body weight. We appreciate the comment of the reviewer because we believe that these new data are of great relevance, since they support the hypothesis that these changes in MKK7 and pJNK are the cause and not the consequence of the development of obesity in our animal model, and that is the reason for not detecting changes in hard endpoints (thermogenesis and body weight) at this specific time point.

7- AgRP-cre p53flox/flox mice display increased food intake only in response to HFD but not to chow diet (fig 3b). This result is of particular interest because it suggests that p53's role in controlling food intake is specific to HFD feeding. Unfortunately, this observation was not investigated any further, but understanding if (and how) p53 responds to excessive caloric intake per se or possible specific nutrient would greatly increase the novelty of this work.

RESPONSE: This is a very good point. It is important to note that the increased food intake was consistently observed in AgRP-cre p53flox/flox mice fed a HFD after 13 weeks (16 week-old) but not before. This means that changes in body weight in our animal model, detected already after 10 weeks of age (Figure 3A) cannot be explained by differences in feeding. To investigate the role of p53 in food intake under HFD, we have now performed a new experiment where

wild type and AgRP-cre p53flox/flox mice fed a HFD (60% fat) were fasted overnight and then refed for 4 hours. As shown in **new Supplementary figure 6A**, mice lacking p53 in AgRP neurons showed a slight but non-significant decrease in fasted-induced hyperphagia.

On the other hand, we have previously found that in mice fed a chow diet, 48 hours of fasting decrease the hypothalamic levels of acetyl p53 whereas these levels are recovered after refeeding (Velasquez DA et al, *Diabetes* 2011, Appendix figure 3, please see below).

[Redacted]

[Redacted]

The changes in p53 expression after fasting/refeeding in mice fed a chow diet also suggest that hypothalamic p53 is not simply altered by an excessive caloric intake, but by nutritional status. Overall, it seems that the interaction between p53 and food intake is quite complex. Considering that this is the first study assessing the metabolic role of hypothalamic p53 and the huge number of biological actions modulated by this transcription factor, we hope that the reviewer will understand that it is practically impossible covering all these aspects in a single paper. Therefore, although we consider that a deep and solid study on the effects of hypothalamic p53 on food intake/nutritional status/food preference/reward (all of them regulated by AgRP neurons) is very interesting, this is beyond of the scope of the current manuscript and will require further study.

8- given the role of inflammation in the control of food intake and energy expenditure, (Garcia-Caceres C, 2013, Solinas 2012) it is surprising that the authors did not measure markers of inflammation in the brain. This is particularly important in studies regarding p53, given its role in stress response.

RESPONSE: We agree with the reviewer that this is an important point. Therefore, we have contacted with Dr. Garcia-Caceres and Prof. Tschöp, now coauthors of the manuscript, and they have measured: a) the number of GFAP+ cells; b) number of primary GFAP+ processes of astrocytes; c) length of primary GFAP+ processes; and the number of Iba1+ cells. As shown in **new Suppl. figures 9A-9F** no differences were detected between control and conditional KOs fed a HFD for 13 weeks in any of these measurements. Moreover, we have also measured protein levels of some inflammatory markers (IL-6 and IL-1 β) in the mediobasal hypothalamus of wild type and conditional KO mice (**new Suppl. figure 9G**) and no differences were found.

9- In Figure 6, the authors turn to ectopic overexpression of p53 using adenovirus expressed p53. Unfortunately, massive overexpression of p53 is likely to drive responses that would be detrimental to the function of cells (such as death and senescence) that may not be reflective of the normal function of endogenous p53, so these overexpression experiments can be very misleading. Again, there is no investigation of p53 activity and how this compared to the activity of p53 when diet is modulated. The use of adriamycin is similarly complicated by the numerous other effects of this drug. Possibly the use of a more selective activator of p53 (like Nutlin) would be more informative.

RESPONSE: We agree with the reviewer that over-expressing a molecule can drive responses that do not necessarily reflect the endogenous function of that molecule. However, we think that to perform gain-and-loss-of-function experiments was an appropriate strategy to demonstrate the specificity of our results. In this case, the over-expression of p53 in the hypothalamic arcuate nucleus leads to a decrease body weight, adiposity and stimulated BAT thermogenesis and energy expenditure, which are the opposite effects to what we observed in mice lacking p53 in AgRP neurons. Thus, overall the former Figure 6, **new Figure 8**, were corroborating that hypothalamic p53 plays an important role in the regulation of energy balance.

Nevertheless, we agree with the reviewer that aspects such as death, senescence and p53 activity deserved to be assessed. Therefore, we have now measured protein levels of markers of apoptosis (caspase 3 and caspase 7) and senescence (PARP) after p53 overexpression in diet-induced obese mice. We found that protein levels of caspase 7 were decreased in the mediobasal hypothalamus of mice where p53 was over-expressed, with no changes detected in caspase 3 or PARP (**new Suppl. Figure 13**). To investigate the potential involvement of death and senescence in the metabolic effects, we also measured these markers of apoptosis and senescence in AgRP-cre p53flox/flox mice fed HFD for 13 weeks. However, we did not find any change in these markers in the conditional KOs (**new Suppl. Figure 9H**). Thus, overall the data indicate that the actions of p53 on body weight are independent of the cell cycle. Indeed, we cannot exclude that hypothalamic p53 might have a role in apoptosis and senescence, but our experiments were performed in relatively young mice that did not present any sign of tumors. Experiments leaving our conditional KOs at long-term would likely be more suitable to unmask whether hypothalamic p53 may affect these aspects. In line with this, we aim to perform a follow-up study to address whether these models have any aging-related alterations and whether the lack of p53 in specific neuronal populations can at least mimic the phenotype of

mice lacking p53 in the whole body. In addition, as measure of p53 activity after the overexpression of p53, we evaluated the level of phosphorylated p53 and acetylated p53 levels in these mice. We found that the phosphorylated form of p53 was significantly increased in the MBH of diet-induced obese mice over-expressing p53 (**new Figure 8B**).

To investigate p53 activity when diet is modulated, we have performed a new experiment where we overexpressed p53 (using viral vectors) in mice fed a chow diet. Under these conditions, we did not find any change in body weight or other metabolic parameters, such as food intake, body composition, BAT thermogenesis or adipocyte size/morphology (**new Suppl. Figure 14**). These new results reinforce the hypothesis that the effects of hypothalamic p53 on body weight and metabolism occur in HFD.

Finally, we understand the doubts of the reviewer regarding the interpretation of the results using the pharmacological stimulation of p53, even though we have previously found that the actions of doxorubicin when administered at peripheral level on body weight are p53-dependent, see below appendix figure 4 from Al Massadi et al, Endocrinology 2016)

[Redacted]

To further explore the specific role of p53 in AgRP neurons avoiding off-target effects, we have now performed a new experiment where we recovered the expression of p53 specifically in AgRP neurons of AgRP-cre p53flox/flox mice fed a HFD injecting AAV-hSyn-DIO-p53. These results are represented in **new Figure 9** and show that, as expected, AgRP-cre p53flox/flox mice gain more weight than their controls when put under a HFD (mice were fed a HFD for 10 weeks before the viral injections), but when these mice were injected with the AAV-hSyn-DIO-p53, they lost weight and fat mass after 4 weeks, which was consistent with a reduction in fat mass, an increase in BAT temperature and higher BAT UCP1 immunostaining. Therefore, these results suggest that p53 specifically over-expressed in AgRP neurons is sufficient to produce a catabolic action.

Minor comments

1- in GTT and ITT absolute values and not percent should be reported.

RESPONSE: we now show GTT and ITT with absolute values as recommended in **new Figures 2C-2D and 3N-3O**.

2- Blood glucose levels in the ITT performed drops quite dramatically. This suggests that the concentrations used are too high and may have masked some phenotype.

RESPONSE: we have repeated the ITT using a lower dose of insulin (0.5 U/kg instead 0.75 U/kg) as shown in **new Figure 3O**, and the results indicate that wild type and conditional KO have a similar insulin sensitivity. We thank the reviewer for this suggestion.

3- The authors claim that only 80% of AgRP or POMC neurones express p53 – it would be more accurate to say only 80% of these cells express levels of p53 that can be detected by immunohistochemistry. I do not think they can conclude (as they do in the discussion) that some of these cells do not express p53.

RESPONSE: We agree with this comment and we have modified the text accordingly. In this regard, it is worth to mention that we now provide data of FACS sorting and single-cell RNA sequencing of AgRP-eGFP neurons and this data show that 31 out of 45 AgRP neurons express p53, indicating that around 69% of AgRP neurons express p53, that is a number close to our previously data shown by IHC (GEO Database repository: GEO Accession: GSE92707) (**new Supplementary figure 4A**).

5- Fig 3 g-h: given the ongoing discussion about how to properly normalise EE data, I would invite the author to report results not normalized on grams as supplementary figure. (Tschoep M. H 2012)

RESPONSE: We are now representing energy expenditure as requested and as it has been represented in other papers from Tschöp's lab (i.e. Fig. 1D from PMID: 28943448). These figures, shown in **new Figures 3G-3H** still show that energy expenditure is decreased in mice lacking p53 in AgRP neurons.

6- Given the role of p53 in cell death and survival, the authors should investigate if AgRP neurons have a different sensitivity to the JNK inhibitor they used. Brain sections after the experiment with the inhibitor displaying similar neuronal population, inflammation and cell death are probably required.

RESPONSE: We agree with the reviewer on this point. We have analyzed brain sections after the administration of the JNK inhibitor and analyzed AgRP fiber density in the ARC and PVH. We found that the number of fibers is significantly increased in both hypothalamic nuclei after

the administration of the JNK inhibitor, indicating that this compound is not damaging these neurons but rather increases their immunostaining (**new Suppl. figure 11**).

We also performed a new experiment using the JNK inhibitor ICV at the same dose than in previous experiments in chow diet fed mice and found that this compound did not cause “per se” any change in body weight, food intake or body composition (**new Suppl. Figures 10A-10E**). Furthermore, we have assessed markers of inflammation, cell death and senescence in these animals. Regarding inflammation, we found that protein levels of IL-1 β but not IL-6 are increased after the icv administration of the JNK inhibitor (**new Suppl. Figure 10F**). Concerning cell death, we found higher protein levels of caspase 3 but not caspase 7 after the icv administration of the JNK inhibitor (**new Suppl. Figure 10G**). Finally, we found no differences in protein levels of PARP between both groups (**new Suppl. Figure 10G**). Despite the increased levels in IL-1 β and caspase 3, the fact that the inhibitor did not itself modify body weight, food intake, adiposity or the number of AgRP fibers suggests that the effects of the JNK inhibitor on mice lacking p53 in AgRP neurons (**new Figure 7**) are not caused by a deleterious action on AgRP neurons.

7- Fig 8: it is unclear if animals were exposed to HFD or to chow diet. However, AgRP-cre p53flox/flox mice display similar food intake on chow diet and even increased food intake on HFD. This would suggest, in a very simplistic manner, a normal or increased sensitivity to Ghrelin. It is possible however that AgRP-cre p53flox/flox are overcompensating ghrelin resistance by more circulating ghrelin levels, which should be measured by ELISA.

RESPONSE: Since it was reported that mice fed a HFD are ghrelin-resistant, for this experiment we used wild type and conditional ko mice fed a chow diet. As suggested by the Reviewer, we have now measured acyl ghrelin levels in these mice and we found a significant increase in mice lacking p53 in AgRP neurons (**new Figure 10G**). Therefore, it seems that these mice are indeed compensating ghrelin resistance by increasing the circulating levels of active ghrelin. We really appreciate this comment as we believe that this is a very relevant aspect to understand this part of the data.

8-It is surprising that the authors did not measure the activity of hypothalamic–pituitary–thyroid axis as it was reported to be involved in the metabolic regulation by JNK (Sabio 2010).

RESPONSE: We have now measured circulating T4 levels in our mice after 13 weeks of HFD to address this point. We found that serum T4 levels were identical in both genotypes (**new Suppl. figure 12**). It is important to note that in the paper cited by the reviewer JNK1 was depleted in the entire murine nervous system, which is completely different to our studies. The different models used are probably explaining the different results on thyroid hormones.

9-It is unclear if the circadian pattern of food intake is affected in AgRP-cre p53flox/flox, especially in response to HFD. Alteration of circadian pattern of feeding has been associated

with obesity (Tureck 2005) and p53 has been reported to interact with the circadian clock (Miki 2013).

RESPONSE: We thank the referee for this interesting suggestion and because of that we have now assessed the circadian pattern of food intake in our mice. We have not found any change in food intake, represented in grams or in kcal, along the circadian cycle in mice lacking p53 in AgRP fed a HFD for 10 weeks, when body weight was already increased compared to their littermates (**new Suppl. figures 6B-6C**). These data suggest that p53 in AgRP neurons is not involved in the interaction of p53 with the circadian clock, and again, we would like to highlight that the increased susceptibility to HFD in mice lacking p53 in AgRP neurons occur independently of feeding, and with the regulation of BAT thermogenesis as the main cause of this phenotype.

10-It is not clear for how many hours animal have been food-deprived before sampling for protein analysis.

RESPONSE: The animals have not been food-deprived before sampling. This explanation will be added to the material and methods.

Reviewer #3 (Remarks to the Author):

Overall this is an interesting hypothesis and novel data implicating p53 in the regulation of energy balance. These observations expand our understanding of the molecular mechanisms contributing to energy balance control by the central nervous system. The investigation of both p53 knock down as well as p53 overexpression strengthens paper as a whole. However, I do feel that the following areas need to be addressed in more detail.

RESPONSE: We would like to thank the Reviewer for the positive and encouraging comments on our manuscript.

Concerns/questions:

1. p53 regulates gene transcription and cell cycle control. Whereas the authors link the loss of p53 to changes in ER stress and JNK activity there is little data or explanation for how p53 activity may mediate these changes observed in the central nervous system. One could argue that loss of p53 will have pleiotropic effects on cell function leading to a range of cellular and physiologic derangements. Given the proposal that the altered physiology seen with p53 deletion is mediated predominantly by the AgRP neuron, studies examining the electrophysiologic function of the AgRP neuron are needed (esp for publication in a high profile journal). In addition, as loss of p53 might lead to developmental changes in AgRP neurons, confirming appropriate AgRP projection targeting/densities is warranted.

RESPONSE: We agree with the Reviewer that these points are very important. Firstly, we have contacted Prof. Peter Kloppenburg to do electrophysiology in control and conditional KO mice fed a HFD to check the electrical activity of AgRP neurons. Despite being more susceptible to HFD, we did not detect any difference in the electrical activity of AgRP neurons in our mouse model (**new Figure 4**). More specifically, spontaneous firing rate, input resistance, spike frequency adaptation, spike threshold and evoked firing of action potentials did not reveal any differences between the AgRP neurons of the two mice lines (**new Figure 4**).

Regarding the lack of changes in the electrical activity of AgRP neurons in a model with strong alterations in body weight, adiposity and energy expenditure, previous studies ablating genes in AgRP neurons show opposed results in this regard. For example, the deletion of ROCK1 in AgRP neurons increases body weight, adiposity and decreases energy expenditure on HFD (PMID: 23885017) but did not change the membrane potential or firing rate of AgRP neurons (Figure 6 of PMID: 23885017). In that study, changes in electrical activity were only clear when mice were challenged with leptin (Figure 6 of PMID: 23885017). In our animal model, we have not measured the electrical activity after an exogenous stimulus because our data clearly show that these mice do not respond to ghrelin (Figure 10). Indeed, other studies have also shown changes in the electrical activity of AgRP neurons in models that show alterations in browning of white fat. An example is the genetic ablation of O-GlcNAc transferase, which inhibits neuronal excitability and promotes white adipose tissue browning, protecting mice against diet-induced obesity and insulin resistance (PMID: 25303527). The reasons for these apparent

discrepancies are likely due to the heterogeneity of AgRP and the existence of different neuronal subsets. In agreement with this heterogeneity, there is a considerable differential gene expression in AgRP neurons after food deprivation (PMID: 26329458). In line with the heterogeneity of AgRP neurons, in the new version of the manuscript, FACS sorting and single-cell RNA sequencing of AgRP-eGFP neurons showed that only 31 out of 45 AgRP neurons express p53 (approx. 69%) (GEO Database repository: GEO Accession: GSE92707) (**new Supplementary figure 4A**). Therefore, it might be possible that some of the recorded neurons would not be affected in our animal model (approx. 31%), masking the “real” effect that p53 may have on AgRP electrical activity.

We also agree with the reviewer that assessing developmental changes in AgRP neurons is important. We have now analyzed brain sections of wild type mice and mice lacking p53 in AgRP neurons fed a HFD for 13 weeks, and measured AgRP fiber density in the ARC and PVH. We found that the number of fibers is identical between the two genotypes, indicating that the lack of p53 is not damaging these neurons, at least when measured at this time point (**new Suppl. figure 7**). In this sense, it is perhaps important to note that mice lacking p53 in the whole body require around 18 weeks to develop tumors, meaning that p53-induced developmental changes in cells is something that requires a considerable amount of time, and therefore, we think that it is not surprising that in our mice we did not find differences in AgRP fiber density at this age. In addition, and as requested also by reviewer 2, we have now investigated whether the deficiency of p53 in AgRP neurons might affect inflammation, apoptosis and senescence, which are all strongly regulated by p53. We found that at the time of the sacrifice, mice lacking p53 in AgRP neurons and fed a HFD for 13 weeks did not show differences in GFAP or Iba1 staining, nor in protein levels of IL-6, IL-1 β , caspase 3, caspase 7 or PARP (**new Suppl. Figures 9A-9H**).

2. The immunohistochemical analysis of p53 and its manipulation is underwhelming (supplemental Figure 3). P53 is poorly visualized and AgRP-Cre activity (as read out by a tomato reporter) is much broader than expected, suggesting ectopic Cre activity in this model. Given that the authors focus their attention on a critical role of p53 in AgRP neurons, additional approaches confirming the loss of p53 only to AgRP neurons are indicated. And again, given the potentially pleiotropic effects of p53 on cell activity, cell counts of AgRP-loxp53 vs AgRP-reporters should be performed to confirm that the number of AgRP cells is similar between groups.

RESPONSE: We agree with the Reviewer that these aspects are important. We have performed additional studies to address this question. As the reviewer may be aware, p53 is widely expressed in most cells. Thus, we investigated which percentage of AgRP neurons that express p53. FACS sorting and single-cell RNA sequencing of AgRP-eGFP neurons showed that 31 out of 45 AgRP neurons express p53 (GEO Database repository: GEO Accession: GSE92707) (**new Suppl. figure 4A**). In addition, we now provide higher magnification photos showing that AgRP-cre p53^{fl/fl} and POMC-cre p53^{fl/fl} have indeed a much lower colocalization of p53 and Tomato/AgRP than their control littermates (**new Suppl. figures 4B and 4C**).

As explained in the previous point, we now measured AgRP fiber density in the ARC and PVH of our mice (**new Suppl. figure 7**) and we did not find differences in cell counts. In addition, we have also quantified the number of AgRP neurons and total number of cells in WT and conditional KOs and found that they are very similar between genotypes (**new Suppl. figure 8**).

3. Figure 1 documents the effects of p53 inhibition by injection of a dominant negative p53 adenoviral vector. Although there is a representative image provided, the authors need to estimate the number of cells transduced by this approach and give some sense of the injection location. Again this is important because the authors have implicated the AgRP neuron as the critical neuronal player in the phenotypes they have uncovered. Are animals with fewer ARC cells transduced less affected by the loss of p53? Are the ARC cells transduced actually expressing AgRP?

RESPONSE: We agree with the reviewer that this information is important. We have now estimated the number of cells transduced by this approach by doing a colocalization of GFP and DAPI. Our new data determine that around 20% of the cells are transduced after the viral injection (**new Suppl. Figure 1**) indicating that even though the virus displays a relatively low infection capacity, it is sufficient to induce metabolic changes. It is important to highlight that the aim of Figure 1 is just to assess whether the ARC is relevant for the actions of p53. In this figure, rats were injected in the ARC with an adenoviral vector inhibiting p53 activity but, of course, with this experiment it is impossible to get solid conclusions. This figure is a preliminary finding simply informing that inhibiting p53 in the ARC causes some effects, and from there on, we generated 2 different mice models targeting the ARC (mice lacking p53 in AgRP and POMC) plus one line targeting the VMH (SF1 neurons). Now, we have also added a new figure overexpressing p53 specifically in AgRP neurons (**new Figure 9**, explained in detail in point 5). In mice lacking p53 in AgRP neurons, we have now done additional measurements after the comments raised by the reviewers, including assessment of density of AgRP fibers (**new Suppl. figure 7**), counting AgRP neurons (**new Suppl. figure 8**), inflammation, apoptosis and senescence (**new Suppl. figure 9**). We think that altogether, the data convincingly indicate that p53 in AgRP neurons regulate energy balance independent of any deleterious action on that population.

4. I am slightly concerned that the bodyweight differences detected following ARC injection of the dominant-negative p53 are really a reflection of an unexpected flattening of weight gain the control group as opposed to increased weight gain in the p53 inhibited group (Figure 1C). Similarly, the phenotypes of the AgRP-cre, lox-p53 mice in Figure 3 is somewhat confusing. Whereas the lines suggest a difference bodyweight on HFD for control and AgRP-cre lox p53 groups, there is a significant flattening of bodyweight gain in the controls at week 16-18 that enhances the effect; interesting, it is only at this point when a clear food intake difference emerges. This needs some explanation, esp if the phenotypic analysis of these mice is performed after the flattening of the controls is observed.

RESPONSE: We appreciate this comment of the Reviewer. Indirect calorimetry, body composition and thermography have been done at week 15 (after 10 weeks on HFD), precisely before the flattening of body weight gain occurs. In our experience, and we have published numerous paper using these approaches (Beiroa D et al. *Diabetes* 2014, Martínez de Morentin PB et al. *Cell Metab* 2014, Imbernon M et al. *Gastroenterology* 2013, Folgueira C et al. *Diabetes* 2016, Martínez-Sánchez N et al *Cell Metab* 2017), after performing these measurements the animals sometimes show changes in body weight. This is something expected, as they are familiarized to live in normal cages in SPF conditions, then they are maintained in cages for the indirect calorimetry system (which are different to the ones they normally live in), and after these analyses, they are again maintained in normal cages; which impacts their stress levels. This means that changing the environment of the animals during these approaches, even though always done in parallel in control and the tested group(s), may affect their level of stress and therefore their body weight, something that is reflected in the graph. Nevertheless, the results show that the conditional KOs were heavier before the analyses (differences in body weight started after 5 weeks on a HFD, meaning 10 week-old mice) and were still heavier after those, meaning that the results are solid and reliable.

5. The authors demonstrate that increased activation of p53 (pharmacologically or genetically) in the ARC can ameliorate diet-induced obesity. These approaches support their hypothesis that p53 activity can drive changes in overall energy balance. These approaches are not limited to manipulating p53 action in the AgRP neuron only; if the authors' central thesis is that p53 action in AgRP neurons is of critical importance, overexpression of active p53 in AgRP neurons specifically would be a more direct test of this question.

RESPONSE: We agree that this is an important question. To further explore the specific role of p53 in AgRP neurons, we have now performed a new experiment where we recovered the expression of p53 specifically in AgRP neurons of AgRP-cre p53flox/flox mice fed a HFD injecting the AAV-hSyn-DIO-p53 virus. These results are represented in **new Figure 9** and show that, as expected, AgRP-cre p53flox/flox mice gain more weight than their controls when put under a HFD (mice were fed a HFD for 10 weeks before the viral injections). When these mice were injected with the AAV-hSyn-DIO-p53 virus, mice lost weight and fat mass after 4 weeks, which was consistent with an increase in BAT temperature and UCP1 expression. Therefore, these results suggest that p53 specifically over-expressed in AgRP neurons is enough to produce a catabolic action. We appreciate the comment of the reviewer since we feel that this new figure corroborates all our previous findings and further strengths the conclusions of the manuscript.

Reviewer #4 (Remarks to the Author):

The manuscript by Quinone and colleagues examines the role of p53 in AgRP neurons in diet-induced obesity (DIO). They report that mice lacking p53 selectively in AgRP, but not POMC or VMN neurons are more susceptible to DIO. This effect is due, in part, to increased energy intake and reduced energy expenditure, and associated with increased hypothalamic JNK activity. In addition, they suggest that pharmacological inhibition or genetic deletion of JNK reverses the obesity phenotype of mice lacking p53 in AgRP neurons. Taken together, the authors conclude that p53 in AgRP neurons is required for normal adaptation against DIO. Overall, the findings are of interest, although several points require additional clarification.

RESPONSE: We would like to thank the Reviewer for the positive and encouraging comments on our manuscript.

1. The increased susceptibility to DIO in mice lacking p53 in AgRP neurons (KO mice), but not POMC or VMN neurons on a HFD is compelling. However, evidence that inhibition of p53 in the ARC increases food intake and body weight in chow-fed rats is relatively small and less convincing. More importantly, how do the authors interpret these findings given that mice lacking p53 in the ARC on a chow-diet do not exhibit a body weight phenotype. Was this study also performed in rats on a HFD? Finally, there is also concern that this viral approach is not selective to the ARC, with expression detected along the walls of the 3V (Figure 1A).

RESPONSE: The Reviewer raises here important points. The aim of Figure 1 (virogenetic knockdown of p53 in the ARC) was just to identify the hypothalamic area where p53 might play a relevant metabolic role. The efficiency of the knockdown is approximately 40% (when measured by p53 protein levels), an average that we often see in our previous studies when manipulating other genes (Porteiro B et al. *Nature Communications* 2017, Imbernon M et al. *Hepatology* 2016, Imbernon M et al. *Molecular Metabolism* 2014, Quiñones M et al. *Molecular Metabolism* 2015). We have now also estimated the number of cells transduced by this approach by doing a colocalization of GFP and DAPI. Our new data determine that around 20% of the cells are transduced after the viral injection (**new Suppl. Figure 1**) indicating that the virus displays a relatively low infection capacity. Therefore, it is not surprising that the effects on body weight are smaller than in mice lacking p53 in AgRP neurons.

Regarding the interpretation of the findings on food intake, we would like to point out that the time when p53 is deleted/inhibited seems to be essential for the phenotype. This was previously demonstrated in a study aiming to investigate the role of p53 in the brown adipose tissue (Al-Massadi O et al, *Endocrinology* 2016). In that paper we showed that mice embryonically lacking p53 in BAT do not show any phenotype, while the inhibition of p53 in the BAT of adult mice increased weight gain. We believe that in the current manuscript, the lack of p53 in AgRP neurons since fetal development does not cause any metabolic alterations under chow diet likely because other compensatory mechanisms occur. However, when the system is forced, and these mice are challenged to HFD, a clear phenotype emerges. This is in fact commonly observed in many different genetically manipulated mice. However, the knockdown of p53 in the ARC of adult animals is sufficient to increase weight gain even though

they are fed a chow diet likely because the lack of compensatory mechanisms replacing p53 activity. Overall, we believe that the results of Figure 1 cannot be directly compared to the conditional KOs, since they are many methodological differences. Figure 1 should be taken as a “fishing expedition”, as it allowed us to identify quickly the importance of the ARC for the actions of p53.

Of course, we control the efficiency of our adenoviral treatments and specially the spreading of adenoviral particles to neighboring hypothalamic nuclei. In our hands, the ARC of rats is targeted with a 70% of success. Also, for each single animal we correlate the efficiency of the injection with the presence of biological effect; in this regard, those animals where the injections were not placed in the ARC did not show biological outcome. In line with this, the inhibition of p53 in the VMH did not cause any change in body weight or food intake (**new Suppl. figures 2A-2C**), indicating that the effects observed after the injection of the viruses in the ARC were specific for this hypothalamic nucleus. It is indeed true that the picture shows some labeling on the walls of the 3V. This is something that we (and other laboratories), often see when injecting viral vectors in the ARC (please find below an appendix figure 5 representing several pictures of GFP staining after injecting the viral vector in the ARC of rats). The reason for this is that because the walls of the 3V, are enriched tanocytes, and these cells act as highly efficient transporters, capturing almost all what they come across. That is why we generated 3 different mice models (mice lacking p53 in AgRP, POMC and SF1 neurons) and now also added a new figure overexpressing p53 specifically in AgRP neurons (**new Figure 9**).

Appendix figure 5. Labeling of GFP after the injection of viral vectors in the ARC.

2. Mice lacking p53 in AgRP neurons exhibit increased body weight when exposed to a HFD. It is recommended that food intake (Fig 3B) be converted to kcal/week instead of grams. In addition, non-fat mass should be presented as grams, rather than as a %BW. This can be misleading, particularly as the KO mice appear to have elevated levels of non-fat mass (Fig 3H).

Are the animals bigger (organ weight, head to anus length) as well as more obese? Given the difference in size of the animals, it is also recommended the multiple linear regression is performed on energy expenditure data (see MMPC website).

RESPONSE: We now show food intake also as kcal/week, body composition (fat mass and non-fat mass) in grams and length of the mice as recommended (**new Supplementary Figure 5**). Regarding the comment of the reviewer on the length of the animals and the energy expenditure, we would like to thank the Reviewer for this comment and we want to apologize because in the former Figure 3H there was a mistake. The reason is that 6 of the conditional KO mice (the six ones that appeared in the right part of the graph showing an extremely high amount of non-fat mass) were represented as body weight in grams instead of non-fat mass. We apologize for this mistake, which we have indeed corrected in the revised version. As suggested by reviewer 2, we are now representing energy expenditure as requested and as it has been represented in other seminal papers from Tschöp's lab (i.e. Fig. 1D from PMID: 28943448). These **new Figures 3G-3H** still show that energy expenditure is decreased in mice lacking p53 in AgRP neurons.

3. The authors also report that the KO mice are more insulin sensitive. This study should be either removed or repeated. Given the blood glucose levels drop below 30mg/dl in both WT and KO mice, this study examines more the response to hypoglycemia, rather than insulin sensitivity. Thus, to test this question, a lower dose of insulin is required.

RESPONSE: We appreciate this comment of the reviewer. We have now repeated the ITT with a lower dose (0.5 U/kg instead 0.75 U/kg) as shown in **new Figure 3O**, and the results have changed because with this dose we found that mice lacking p53 in AgRP neurons show a similar response to insulin as control mice. We thank the reviewer for this suggestion as this is an important aspect. To note, we now represent GTT and ITT in absolute values as requested by reviewer 2.

4. In the subsequent studies, the authors examine the effects on ER stress and JNK in WT and KO mice on a HFD. A subtle, but important point that requires attention and is important in the interpretation, is that previous evidence suggests that ER stress and JNK is increased in HFD-fed relative to chow-fed WT mice. Thus, statements throughout the ms are suggested to be revised accordingly. For example (line 222) to read ... these results suggest that "further" activation.... In addition, studies using the ER stress reliever (TUDCA; Fig. 4) or the JNK inhibitor SP-600125 (Fig. 5) may also benefit from a chow-fed control group. Overall, the findings on body weight from both of these studies are a little underwhelming, particularly as the control-saline-treated group lose considerable body weight. Moreover, given that HFD induces ER stress and JNK, based on the previous literature, one would predict that these treatments would induce weight loss in HFD WT mice, but the weight loss would have been exacerbated in the KO mice. This warrants further discussion. Food intake is also best presented as mean daily energy intake, rather than cumulative energy intake.

RESPONSE: We thank the reviewer for this comment and will revise the text accordingly. Regarding former figures 4 and 5 (**new figures 6 and 8**), we treated mice fed a HFD with TUDCA (a chemical chaperone that reduces ER stress) and the JNK inhibitor SP-600125 precisely because as the reviewer states, both ER stress and JNK are induced in HFD; and second, because mice lacking p53 in AgRP neurons only show a phenotype when fed a HFD. Lean animals already have low levels of ER stress or active JNK and therefore, from our point of view, it is not clear the rationale for inhibiting ER stress and JNK in lean animals. As a matter of fact, in one of our previous studies we already treated mice ICV with TUDCA in mice fed a chow diet and we did not detect differences in food intake, body weight or adiposity (please see below figure taken from Supplementary figure 6, Schneeberger M et al, *Cell* 2013). We found similar results when we infused TUDCA at central level in rats fed a chow diet (Figure 2A, Contreras C et al, *Diabetes* 2017).

[Redacted]

Regarding the administration of the JNK inhibitor SP-600125 in mice fed a chow diet, we have now done this experiment and found that its ICV injection (1.5 ug/mouse for 6 days), meaning same dose and time of treatment as in the former Figure 5 (**new Figure 7**) did not modify food intake, body weight or body composition (**new Suppl. Figures 10A-10E**).

Regarding the weight loss of control groups during the ICV treatments, this is something that we normally see in our experiments performed in obese mice. Mice fed a HFD are very sensitive to surgeries, and the normal response of these mice under these conditions is to show a decrease in body weight. However, since we have all the appropriate groups treated at the same time (in this case control and conditional KO mice treated either with ICV vehicle or TUDCA in **new Figure 5**, and ICV vehicle or SP-600125 in **new Figure 7**) the different groups can be compared. And in this case, the effects of the compounds are clearly different in control mice or mice lacking p53 in AgRP neurons fed under the same diet.

We agree with the Reviewer that the effects of the inhibition of ER stress and JNK in rodents fed a HFD are well known. However, these results are highly dependent on the dose used. For example, TUDCA is normally given ip at 500 mg/kg (Ozcan U et al, *Science* 2006), which means that for a mouse of 40g the correspondent dose is 20 mg. In a previous study we used 10

ug/day in rats fed a HFD (Contreras C et al, *Diabetes* 2017) and we found that ICV TUDCA was able to decrease body weight. In the present work, we use a lower dose of TUDCA (5 ug/mouse) and we are not able to detect changes in body weight or food intake in control mice fed a HFD after 1 week. Our purpose here was to demonstrate that using subeffective doses of TUDCA and SP-600125, mice lacking p53 in AgRP neurons were sensitive to the inhibition of ER stress and JNK than their control littermates. Indeed, we cannot exclude that increasing the length of the treatment might be sufficient for detecting changes in body weight. Finally, food intake is now also represented as suggested (**new Figure 5E and Figure 7C**).

5. The color scheme for the Legend in Figure 5L needs to be revised.

RESPONSE: We thank the reviewer for detecting this mistake. We have now revised this legend (**new Figure 7K**).

6. To determine whether pharmacological activation of p53 in the ARC is sufficient to attenuate food intake and body weight gain, animals receive Adriamycin (a drug activating p53) directly into the ARC using osmotic minipumps. There are major concerns with this approach, given that although the cannula is directed to the ARC, the minipump delivers 1ul/hr (24ul/d) and will not only leak into adjacent hypothalamic nuclei, but into the 3V. Thus, the p53 can act anywhere throughout the brain. The authors provide data that delivery into the VMH infusion of Adriamycin does not have any feeding or body weight effects, although this cannot be fully interpreted as the authors only examine food intake and body weight over 5 days, a time when there was no effect following ARC administration.

RESPONSE: This is a good comment from the Reviewer. We acknowledge that the chronic direct administration of Adriamycin in each hypothalamic area might not be 100% specific but we have used this approach in previous articles injecting other compounds in specific hypothalamic nuclei (Martinez de Morentin, *Cell Metab* 2014, Martins L et al. *Cell Reports* 2016, Contreras C et al. *Diabetes* 2017, Imbernon M et al. *Gastroenterology* 2013; Martinez-Sanchez, *Cell Metab* 2017) and when we injected those compounds in other hypothalamic areas the effects were not reproduced. The results of this pharmacological stimulation of p53 agree with the genetic disruption of p53. However, since other reviewers also raised concerns about the specificity of these data and the use of this compound, we have decided to perform a new experiment where we recovered the expression of p53 specifically in AgRP neurons of AgRP-cre p53flox/flox mice fed a HFD injecting the AAV-hSyn-DIO-p53 virus. These results are represented in **new Figure 9** and show that, as expected, AgRP-cre p53flox/flox mice gain more weight than their controls when put under a HFD (mice were fed a HFD for 10 weeks before the viral injections). When these mice were injected with the AAV-hSyn-DIO-p53 virus, mice lost weight and fat mass after 4 weeks, which was consistent with an increase in BAT temperature and UCP1 expression. Therefore, these results suggest that p53 specifically over-expressed in AgRP neurons is enough to promote a catabolic effect. We appreciate the

comment of the reviewers since we feel that this new figure corroborates all our previous findings and strengthens the conclusions of the manuscript.

7. The authors provide evidence that in mice lacking p53 in AgRP neurons exhibit a blunted response to ghrelin. Are these in chow or HFD-fed mice? Can the authors discuss how this observation might contribute to the increased susceptibility to DIO?

RESPONSE: We did this experiment on chow diet because is when the ability of ghrelin to increase food intake and body weight is intact. In DIO animals however there is a resistance to the central orexigenic action of ghrelin, reviewed in (Zigman JM et al *Trends Endocrinol Metab.* 2016 May;27(5):348, Cui H et al *Nat Rev Endocrinol.* 2017 Jun;13(6):338-351). Therefore, to compare the orexigenic and adipogenic action of ghrelin between wild type and conditional KOs we did this experiment in chow diet.

Since this experiment was performed in mice fed a chow diet it is difficult to know precisely how this might contribute to the increased susceptibility to DIO of mice lacking p53 in AgRP neurons. One possibility is that HFD engages an alternative ghrelin-induced hedonic pathway that bypasses homeostatic mechanisms –namely AgRP neurons– to induce adiposity contributing to the obesogenic phenotype (Dennis RG et al 2015). However, we cannot rule out that other hormones and/or factors are also contributing to this increased susceptibility to DIO.

What we have done, following the request of reviewer 2, was to measure acyl ghrelin levels in these mice and we found a significant increase in mice lacking p53 in AgRP neurons (**new Figure 10G**). Therefore, it seems that these mice are overcompensating ghrelin resistance (in chow diet) by increasing the circulating levels of active ghrelin. We really appreciate this comment as we believe that this is a very relevant aspect to understand this part of the results.

Reviewer #2 (Remarks to the Author):

The authors have addressed many of the reviewers' comments and the study is much stronger. There are a few points where the authors may wish to be more cautious with the conclusions.

1. In general, the differences in metabolic cages are seen after differences in body weight, so it is still possible that the changes in metabolism are due to, rather than the cause of, body weight increase. Figure 6 shows no real change in energy expenditure before the onset of obesity, so it might be prudent to avoid stating that the mice are leaner because of increased BAT thermogenesis (even though this is likely to be correct).
2. While the phenotype is described as being independent of food intake, Figure 3 shows that the p53 ko mice do eat somewhat more from week 6-9. While this is not significant at each time point, cumulative intake may differ. Again, it could be better not to make such strong statements.
3. Is it possible that the JNK inhibitor has a more profound effect on the p53 ko neurons?
4. A slight trend in increased food intake is already evident in weeks 1-3 (Figure 6), which may contribute to weight gain despite not being statistically significant at any single time point.
5. The GTT in Figure 3O shows no difference, despite differences in body weight. The KO mice also seem to be more insulin sensitive. Could this be discussed a bit more?

Reviewer #3 (Remarks to the Author):

Nogueiras and colleagues have submitted a revised version of their manuscript entitled "p53 in AgRP neurons is required for protection against diet-induced obesity via JNK1". The research team is to be complimented on the extent of their additional efforts. My previous concerns have been addressed by new data examining the electrophysiologic properties of Agrp neurons lacking p53 and both the cell number and axonal projections of these neurons. That these properties are not altered by deletion of p53 supports the hypothesis that p53 function (presumably through JNK) is an important regulator of energy balance control. Moreover, the re-expression of p53 specifically in the Agrp neurons of Agrp-Cre flox p53 mice and the subsequent metabolic improvements significantly strengthens the paper.

Reviewer #4 (Remarks to the Author):

The authors have addressed the Reviewers concerns in a thorough and satisfactory manner.

Response to Reviewer #2

1. In general, the differences in metabolic cages are seen after differences in body weight, so it is still possible that the changes in metabolism are due to, rather than the cause of, body weight increase. Figure 6 shows no real change in energy expenditure before the onset of obesity, so it might be prudent to avoid stating that the mice are leaner because of increased BAT thermogenesis (even though this is likely to be correct).

REPLY: We have now changed the text accordingly (basically abstract and discussion) and have avoided the state that manipulation of p53 in the hypothalamus is the direct cause of changes in BAT activity

2. While the phenotype is described as being independent of food intake, Figure 3 shows that the p53 ko mice do eat somewhat more from week 6-9. While this is not significant at each time point, cumulative intake may differ. Again, it could be better not to make such strong statements.

REPLY: During the previous revision, we already avoided to mention that changes in body weight found in AgRP-Cre^{p53loxP/loxP} mice were independent of feeding. As a matter of fact, we wrote in the discussion that these mice had slight hyperphagia. We think that all these strong statements were already corrected in the previous version.

3. Is it possible that the JNK inhibitor has a more profound effect on the p53 ko neurons?

REPLY: With our present data we found that the used dose of the JNK inhibitor increased protein levels of IL-1 β and caspase 3 in the mediobasal hypothalamus of wild type mice (Supplementary Figures 10F-10G) and that the JNK inhibitor did not negatively affect AgRP fiber density in the ARC and PVH of wild type mice (Supplementary Figure 11), indicating that this compound is not damaging these neurons. Whether the compound may have more profound effects in AgRP neurons of conditional KOs on some biochemical pathways has not been evaluated, but the fact that this compound reverses the obese phenotype of mice lacking p53 in AgRP neurons (which have higher protein levels of JNK in the mediobasal hypothalamus) suggests that some molecular mechanism in these neurons should be affected

4. A slight trend in increased food intake is already evident in weeks 1-3 (Figure 6), which may contribute to weight gain despite not being statistically significant at any single time point.

REPLY: We agree with the reviewer that there is a tendency of increased food intake in conditional KO mice, therefore we have modified the text accordingly (page 12).

5. The GTT in Figure 3O shows no difference, despite differences in body weight. The KO mice also seem to be more insulin sensitive. Could this be discussed a bit more?

REPLY: Conditional KO mice show a significant difference only in one time point (30 min) while the other time point only show a tendency. Thus, with the present data, we believe that to say that these mice are more insulin sensitive would be an overstatement. That being said, it is true that since mice lacking p53

in AgRP neurons are more obese than their littermates, one could expect a worsened glucose tolerance and lower insulin sensitivity, which is definitely not the case. AgRP neurons have been shown to control glucose homeostasis through their actions on several tissues, including muscle or pancreas. Thereby, it might be possible that p53 in AgRP neurons differentially affect organs playing a key role in the regulation of glucose homeostasis, favouring insulin sensitivity despite the obese phenotype of mice lacking p53 in AgRP neurons.